
# New particle formation at urban and high-altitude remote sites in the south-eastern Iberian Peninsula

Juan Andrés Casquero-Vera[1,2], Hassan Lyamani[1,2], Lubna Dada[3], Simo Hakala[3], Pauli Paasonen[3], Roberto Román[4], Roberto Fraile[5], Tuukka Petäjä[3], Francisco José Olmo-Reyes[1,2], Lucas Alados-Arboledas[1,2]

[1]Andalusian Institute for Earth System Research (IISTA-CEAMA), University of Granada, Autonomous Government of Andalusia, Granada, Spain.
[2]Department of Applied Physics, University of Granada, Granada, Spain
[3]Institute for Atmospheric and Earth System Research (INAR)/Physics, Faculty of Science, University of Helsinki, Helsinki, Finland
[4]Grupo de Óptica Atmosférica (GOA), Universidad de Valladolid, Valladolid, 47011, Spain
[5]Department of Physics, University of Leon, Leon, Spain

*Correspondence to*: Juan Andrés Casquero Vera (casquero@ugr.es)

**Abstract.** A substantial fraction of the atmospheric aerosols originates from secondary new particle formation (NPF), where atmospheric vapours are transformed into particles that subsequently grow to larger sizes, affecting human health and the climate. In this study, we investigate aerosol size distributions at two stations located close to each other (~20 km), but at different altitudes: urban (UGR; 680 m a.s.l.) and high-altitude remote (SNS; 2500 m a.s.l.) site, both in the area of Granada, Spain, and part of AGORA observatory (Andalusian Global ObseRvatory of the Atmosphere). The analysis shows a significant contribution of nucleation mode aerosol particles to the total aerosol number concentration at both sites, with a contribution of 47% and 48% at SNS and UGR, respectively. Due to the important contribution of NPF events to the total aerosol number concentrations and their high occurrence frequency (>70%) during the study period, a detailed analysis of NPF events is done in order to get insight into the possible mechanisms and processes involved in NPF events at these contrastive sites. At SNS, NPF is found to be associated with the transport of gaseous precursors from lower altitudes by orographic buoyant upward flows. However, NPF events at SNS site are always observed from the smallest measured sizes of the aerosol size distribution (4 nm), implying that NPF takes place in or in the vicinity of the high-altitude SNS station rather than transported from lower altitudes. Although NPF events at the mountain site seem to be connected with those occurring at the urban site, growth rates (GR) at SNS are higher than those at UGR site ($GR_{7-25}$ of 6.9 and 4.5 nm h$^{-1}$ and $GR_{4-7}$ of 4.1 and 3.6 nm h$^{-1}$ at SNS and UGR, respectively). This fact could have a special importance on the production of cloud condensation nuclei (CCN) and therefore on cloud formations which may affect regional/global climate, since larger GR at mountain sites could be translated to larger survival probability of NPF particles to reach CCN sizes, due to shorter time needed for the growth. The analysis of sulfuric acid ($H_2SO_4$) shows that the contribution of $H_2SO_4$ is able to explain a minimal fraction contribution to the observed GRs at both sites (<1% and <10% for 7-25 and 4-7 nm size range, respectively), indicating that other condensing vapours are



responsible for the majority of particle growth, as well as the differing growth rates between the two sites. Results also show that the condensation sink (CS) does not play a relevant role in NPF processes at both sites and points to the availability of volatile organic compounds (VOCs) as one of the main factors controlling the NPF events at both sites. Finally, a closer analysis of the NPF events that were observed at SNS site during a Saharan dust episode that occurred during the field campaign

was carried out, evidencing the role of $TiO_2$ and $F_2O_3$ together with VOCs in promoting new particle formation during this dust intrusion event. Although further investigation is needed to improve our understanding in this topic, this result suggests that climate effects of mineral dust and NPF are not disconnected from each other as it was commonly thought. Therefore, since mineral dust contributes to a major fraction of the global aerosol mass load, dust-NPF interaction should be taken into account in global aerosol-climate modelling for better climate change prediction.

**1 Introduction**

The formation of new atmospheric aerosol particles, and their subsequent growth, commonly known as New Particle Formation (NPF) events, has a substantial contribution to aerosol particle number concentration and thus affect the climate via aerosol-cloud interactions (Kerminen et al., 2018). Kulmala et al. (2014) described NPF events as a five-step process: (1) chemical reactions in the gas phase producing low-volatility vapour(s), (2) clustering, (3) nucleation or barrier-less nucleation, (4)

activation of clusters with a second group of vapours, and (5) subsequent multi-component condensational growth of nucleated particles to larger sizes. However, despite the advancement in theoretical knowledge of NPF steps, large discrepancies have been found between the expected and observed properties of NPF under atmospheric conditions (Chu et al., 2019; Kulmala et al., 2017; Nieminen et al., 2018). Thus, more NPF studies in different environments and conditions are still needed for better understanding of the NPF processes.

Sulfuric acid ($H_2SO_4$) has been commonly considered as one of the main precursors for aerosol nucleation and growth due to its low vapour pressure. However, $H_2SO_4$ alone is not sufficient to explain the observed NPF under various ambient conditions (Kulmala et al., 2004; Kulmala and Kerminen, 2008; Zhang et al., 2012). Earlier studies indicated that ammonia ($NH_3$) enhances aerosol nucleation, but recent laboratory and theoretical studies suggest that amines and highly oxygenated molecules (HOMs) play vital roles in enhancing nucleation and promoting the initial growth of newly formed particles in the atmosphere

(e.g., Kulmala et al., 2013; Schobesberger et al., 2013; Tröstl et al., 2016; Ehn et al., 2014).

New particle formation accounts for approximately 50% of the aerosol number concentration production in the troposphere (Merikanto et al., 2009; Spracklen et al., 2010) but frequency, intensity or duration of NPF events are highly variable, being the prediction of NPF events a current challenge of large interest. It is known that various factors favour the occurrence of NPF events. Precursor gases, concentration of pre-existing aerosol, meteorological variables and solar radiation are some of

these key factors that govern NPF events (Dada et al., 2017). However, current knowledge about NPF processes remains poor, especially at high altitudes, i.e. above 1000 m (Rose et al., 2015) and at megacities (Kulmala et al., 2017; Yao et al., 2018),



and a more profound understanding of the mechanisms and precursor gases involved in nucleation and particle growth is currently required, especially to improve the accuracy of climate models. Thus, in order to improve the actual understanding of NPF events, it is necessary to characterize NPF events at different environments with distinct mixture of anthropogenic and natural precursor gas emissions and meteorological conditions.

Aerosol also plays an important role in the process of cloud formation by acting as cloud condensation nuclei (CCN) (e.g., Pierce and Adams, 2009; Spracklen et al., 2008). The aerosol-cloud interaction depends, mainly, on the water vapour supersaturation and the chemical composition and size distribution of the aerosol particles. Typically, the size at which aerosols activate as CCN ranges from 50 to 150 nm (Kerminen et al., 2012), and the NPF events are one of the main processes producing aerosols in these sizes. In this sense, according to Merikanto et al. (2009), 45% of global low-level cloud CCN are originated
from nucleation, and 35% of these CCN are formed in the free and upper troposphere.

Due to the role of NPF in the production of CCN, it is of special interest to characterise NPF in the free troposphere (FT) and its vertical distribution. However, only a few studies have focused on the vertical distribution of these events (e.g., Carnerero et al., 2018; Komppula et al., 2003) and the existing observations of NPF at high altitudes provide contradictory results. For example, Crumeyrolle et al. (2010) showed that the occurrence of NPF events was limited to the planetary boundary layer
(PBL), while Hamburger et al. (2011) have reported the presence of high concentrations of ultrafine particles in the upper FT, and Boulon et al. (2011) found that the nucleation processes were more frequent in a high-altitude site than in the PBL, due to enhanced ion contribution to NPF at high altitudes. Also, the occurrence of the NPF in the FT has been reported to be tightly connected with the strength of boundary-layer influence at the site, together with global radiation (Bianchi et al., 2016; Tröstl et al., 2016). Furthermore, Foucart et al. (2018) suggested that the turbulent mixing between the boundary layer and the FT
could enhance the NPF events at the interface between these layers. Thus, NPF studies, especially at high altitudes where environmental conditions favour cloud formation, are still needed for improving our understanding of the NPF processes.

Long-term observations at high altitudes are difficult to perform due to the lack of availability of suitable sites and due to adverse meteorological conditions. Also, interpretation of data from high altitude stations must be done with care, as a station may not always be representative of the altitude it lays at due to frequent valley winds and topographic effects (Sellegri et al.,
2019). In the last years, some studies of NPF were conducted in high altitude sites (Boulon et al., 2011; García et al., 2014; Rose et al., 2015), but better understanding of the NPF processes in high altitude sites is still needed. Event frequency, growth and formation rates of NPF events at high altitude stations differ greatly from each other, and new insights in this field are necessary (Sellegri et al., 2019).

In this study, we investigate aerosol size distributions in the size range 4-500 nm measured simultaneously during an intensive
summer campaign at two stations located close to each other (~20 km), but at different altitudes: urban (680 m a.s.l.) and high-altitude remote (2500 m a.s.l.) site, in southern Spain. We also analysed and discussed the occurrence frequency, characteristics and factors that promote/inhibit NPF processes at these contrastive sites during this campaign. The role of precursor gases



such as sulfuric acid and volatile organic compounds and their contribution to the formation and growth of aerosol particles at these two different environments are explored. Finally, a closer analysis of the NPF events that were observed during a Saharan dust episode that occurred during the field campaign was carried out in order to get insight into the processes and constituent components that lead to the formation of new particles during this dust intrusion event.

## 2 Measurements and methods

### 2.1 Measurement sites and instrumentation

Measurements were conducted in the framework of the SLOPE II campaign (Sierra Nevada Lidar AerOsol Profiling Experiment II). The SLOPE II campaign was developed along summer 2017 (May - September) combining active and passive remote sensing with in-situ measurements at two different altitude sites in the northern slope of Sierra Nevada mountain range (Granada, Spain). The used measurements were obtained during an intensive campaign from 23 June to 24 July at Granada urban station (UGR) and Sierra Nevada remote station (SNS), both sites included in the AGORA observatory (Andalusian Global ObseRvatory of the Atmosphere). During this intensive campaign, these two permanent stations were also equipped with extra equipment, monitoring a large variety of aerosol properties, gaseous species and radiation parameters.

The UGR urban station is located in the "Andalusian Institute for Earth System Research" (IISTA-CEAMA), in Granada city, Spain (37.16° N, 3.61° W, 680 m a.s.l.). Granada is a Spanish city located in the south-eastern Iberian Peninsula, in a natural basin surrounded by mountains, especially by the mountain range of Sierra Nevada, with peaks up to 3300 m a.s.l., showing a Mediterranean-continental climate. It is a medium-size city with a population of about 250.000 inhabitants. Granada is a non-industrialized city and one of the Spanish cities that suffers from pollution problems (Casquero-Vera et al., 2019). The main aerosol sources in Granada city are domestic heating based on fuel oil combustion and biomass burning in winter, Saharan desert dust in summer, and heavy traffic throughout the year (Lyamani et al., 2010, 2011; Patrón et al., 2017; Titos et al., 2012, 2017).

The SNS remote station (37.10°N, 3.39°W, 2500 m a.s.l.) is locate in the northern slope of Sierra Nevada mountain range, ~5 km northwest of the Veleta summit (3398 m a.s.l.) and ~20 km southeast of Granada city (Fig. 1). Due to its location with respect to the valley, prevalent wind directions are mainly from westerly and southerly directions, that favours the transport of pollutants from lower altitudes, mainly from the Granada city, to SNS station. Thus, this station provides measurements, which can represent the south-western European free troposphere as well as give us insight on the impact of the aerosol particles transported from lower altitudes, especially from Granada, the nearest city, and its surroundings.

*[Figure 1]*

Both stations are equipped with a large number of instruments for measuring aerosol properties as well as radiation and meteorological parameters (wind speed and direction, temperature, relative humidity, pressure and precipitation). Aerosol





properties and meteorological as well as radiation parameters were measured with the same instrumentation model at both stations, following same calibration procedure and data quality criteria. Focusing on the measurements used in this work, aerosol particle number size distributions from 4 to 500 nm were measured every 5-min by nano-SMPS and SMPS (TSI model 3938), both composed of an electrostatic classifier (TSI 3082) and a Condensation Particle Counter (CPC; TSI 3772 for SMPS

and TSI 3775 for nano-SMPS) using aerosol flow rates of 1.0 lpm (SMPS) and 1.5 lpm (nano-SPMPS) and sheath flow of 5.0 lpm for both systems. The quality of the SMPS measurements was checked for flow rates, RH, 203 nm PSL calibration and in-situ intercomparison (ACTRIS Round Robin Tour), following the ACTRIS and GAW recommendations (Wiedensohler et al., 2012, 2018). Following SMPS calibration procedures, uncertainty in the measured particle size distribution is within 10% and 20% for the size ranges 20-200 nm and 200-800 nm, respectively (Wiedensohler et al., 2018).

UV-B (280-320 nm) was monitored using UVB-1 (Yankee Environmental Sytems) pyranometers, wind speed and direction were measured with RM-Young 05102 anemometers and temperature and humidity with Rotronic HygroClip 2 (HC2) probes. Particle number size distribution in the aerodynamic diameter range of 0.5–20 μm was measured with an Aerodynamic Particle Sizer (APS; TSI model 3321). Aerosol 24-h $PM_{10}$ samples were collected on quartz fiber filters by means of high-volume sampler with a flow rate of 30 $m^3$ $h^{-1}$. The filters were conditioned and treated pre- and post-sampling and then a complete

chemical analysis was performed for all samples at Institute of Environmental Assessment and Water Research (IDAEA-CSIC, Barcelona, Spain) following the procedure of Querol et al. (2001).

Pollutant gases (CO, NO, $NO_2$, $NO_x$, $SO_2$ and $O_3$) data from Ciudad Deportiva air quality station were used. This station is an urban background station located at ~3 km from UGR station. These data are provided by the Junta de Andalucía (http://www.juntadeandalucia.es/medioambiente), following the requirements of the European AQ directives. At SNS station,

concentrations of trace gases (CO, NO, $NO_2$, $NO_x$, $SO_2$ and $O_3$) were monitored, following the European AQ directives requirements, by a mobile cabin of the University of Évora. Finally, ambient air samples were collected at both sites on TENAX, C18 and DNPH cartridges with a maximum 12-h time resolution by low-volume pumps. Volatile organic compounds retained on solid-phase filters are then chemically analysed using gas chromatography or liquid chromatography coupled to mass spectroscopy techniques (Borrás, 2013).

**2.2 Data analysis**

In this study, the measured 5-min particle number concentration from the SMPS is divided into three diameter ranges: nucleation mode from 4 to 25 nm ($N_{4-25}$), Aitken mode from 25 to 100 nm ($N_{25-100}$) and accumulation mode from 100 to 500 nm ($N_{100-500}$). In addition, the total number concentration is calculated from the whole 4 to 500 nm range ($N_{Tot}$).

The classification of NPF event days was done by visual interpretation of the daily particle number size distribution data

according to the guidelines presented by Dal Maso et al. (2005). According to this classification criteria, days are classified and separated into four groups: event (E), non-event (NE), undefined (UN) and bad-data days (BD). (1) "E" days are days





during which sub-25 nm particle formation and their consequent growth are observed; (2) "NE" days are days on which neither new growing modes or production of sub-25 nm particles are observed; (3) "UN" days are the days which do not fit either of the previous classes; (4) "BD" days are the days during which data are not valid or inexistent. In addition, event days are separated into two different groups: class I and II events. NPF events are classified as class I events when the NPF growth rate

retrieval is possible from 7 to 25 nm, and class II when it is not possible.

In this study, the automatic DO-FIT algorithm (Hussein et al., 2005) is used to describe the measured particle number size distributions by fitting multiple lognormal distributions to the measured data. The geometric mean diameters of the fitted distributions are then used for calculating particle growth rates during NPF events. This is done by calculating the slope of the linear fit to the geometric mean diameters as a function of time, which were identified to represent the growing particle mode

formed in an NPF event. Thus, the growth rate (GR) is obtained as:

$$GR = \frac{dD_p}{dt} = \frac{\Delta D_p}{\Delta t} \qquad \text{Eq. (1)}$$

where $D_p$ is the representative diameter of the NPF mode at time $t$. In this work, the growth rates in the range 4-7 ($GR_{4-7}$) and 7-25 nm ($GR_{7-25}$) were calculated. The uncertainties in the calculated GRs was estimated on 19% and 8% for the 3-7 and 7-20 nm size ranges (Yli-Juuti et al., 2011).

The formation rate ($J_{D_p}$) is calculated following the methodology described by Kulmala et al. (2012):

$$J_{D_p} = \frac{dN_{D_p}}{dt} + CoagS_{D_p} \cdot N_{D_p} + \frac{GR}{\Delta D_p} \cdot N_{D_p} \qquad \text{Eq. (2)}$$

where the first term on the right hand side represents the observed change in particle concentration ($N_{D_p}$: particle number concentration for a determined diameter size); the second term describes the loss of particles due to coagulation with larger aerosol particles; and the third term considers the growth out of the considered size range. In this study, the formation rates at

4 ($J_4$) and 7 nm ($J_7$) were calculated.

The condensation sink (CS) describes how rapidly vapour molecules will condense onto pre-existing aerosols. CS is dependent on the effective surface area of the pre-existing particle size distribution (Kulmala et al., 2012). Accordingly, the CS is calculated from each size distribution as:

$$CS = 2\pi D \int_{D_{min}}^{D_{max}} D_p \, \beta_M \, N_{D_p} \, dD_p = 2\pi D \sum_{D_p} D_p \, \beta_M \, N_{D_p} \qquad \text{Eq. (3)}$$

where $D$ is the diffusion coefficient of condensable vapour, that is assumed to be sulfuric acid, and $\beta_M$ is the transitional correction factor (Fuchs and Sutugin, 1971) which dependent on the mean free path of vapour molecules and aerosol diameter (e.g., Kulmala et al., 2001; Pirjola et al., 1999).



In order to quantify the contribution of $H_2SO_4$ to new particle formation and growth, we determine the concentration of condensable vapour required for a growth rate of 1 nm $h^{-1}$ ($C_v$) following Nieminen et al. (2010) procedure. For better estimation of condensable vapour concentration, $C_v$ was retrieved by using the mass and density of the sulfuric acid-water-mixture (Kurtén et al., 2007) taking into account the differences of temperature and RH between SNS and UGR station.

Since concentration of $H_2SO_4$ was not measured at either stations, a proxy of $H_2SO_4$ concentration is used in order to define whether $H_2SO_4$ is the precursor vapour responsible for NPF at both stations and if so to link the new particle formation rates with the $H_2SO_4$ concentrations The proxy calculation was first proposed by Petäjä et al. (2009) which derives $H_2SO_4$ concentration from its formation through $SO_2$ concentration in the presence of global radiation, UV-B or OH and its loss to CS. To estimate the $H_2SO_4$ concentration for our two stations, we used the following formula (Petäjä et al., 2009):

$$H_2SO_4 = k \cdot \frac{SO_2 \cdot UVB}{CS} \qquad\qquad\qquad\qquad\qquad\qquad\qquad\qquad Eq.\ (4)$$

where $SO_2$ is sulfur dioxide concentration, UV-B is the radiation with wavelength from 280 to 320 nm and $k$ is a scaling factor. For this work, we use the $k$ value reported by Petaja et al. (2009) based on measurements in the Hyytiala SMEAR II station ($k = 8.4 \times 10^{-7} \cdot UVB^{-0.68} \ [m^2 W^{-1} s^{-1}]$). The choice of $k$ value will have an influence on the absolute value of the $H_2SO_4$ concentrations, but not on the relative variability. Then, the particle growth rate by sulfuric acid is calculated directly as the

ratio of $H_2SO_4$ and the concentration of condensable vapours required for a growth rate of 1 nm $h^{-1}$ ($C_v$).

Finally, the dimensionless survival probability parameter ($P$) proposed by Kulmala et al. (2017) was calculated as:

$$P = CS'/GR' \qquad\qquad\qquad\qquad\qquad\qquad\qquad\qquad\qquad\qquad Eq.\ (5)$$

where $CS' = CS / (10^{-4} \ s^{-1})$ and $GR' = GR_{7-25}/ (1 \ nm \ h^{-1})$. The used CS and GR values were calculated with the methods described previously. The larger the survival parameter is, the smaller percentage of newly formed particles will survive to

greater sizes. Values of $P$ smaller than 50 are typically required for NPF occurrence in clean and moderately polluted environments, although higher values of $P$ (up to 200) are observed for NPF events in highly polluted atmospheres (Kulmala et al., 2017).

## 3 Results and discussion

### 3.1 Atmospheric aerosol number concentrations

During the intensive SLOPE II campaign, 32 days of coincident aerosol number concentration measurements at SNS and UGR station were recorded. A statistical summary of daily mean particle number concentrations in different size ranges ($N_{4–25}$, $N_{25–100}$, $N_{100–500}$ and $N_{Tot}$) at SNS and UGR station are presented in Table 1. As expected, the average total particle number concentration during the whole campaign period at UGR urban station was clearly 2 folds larger (mean value of $8.4 \times 10^3$ cm$^{-3}$) than at the remote SNS station ($3.4 \times 10^3$ cm$^{-3}$) due to a large contribution from local anthropogenic emissions in the urban





environment. Total aerosol number concentration at UGR is comparable to those reported for other European and Spanish urban sites. For example, Birmili et al. (2016) reported aerosol number concentrations ranging from 5.0 to $9.6 \times 10^3$ cm$^{-3}$ in 7 different urban background sites in Germany. Nevertheless, aerosol number concentrations obtained in UGR are slightly higher than those observed by Gómez-Moreno et al. (2011) ($N_{13-600} \sim 7.5 \times 10^3$ cm$^{-3}$) for July in an urban background site in Madrid.

However, Pey et al. (2010) reported $N_{13-800}$ of $16 \times 10^3$ cm$^{-3}$ on July and Pérez et al. (2010) reported $N_{5-1000}$ of $24 \times 10^3$ cm$^{-3}$ for the period July-November, both for an urban background site in Barcelona, which are slightly higher concentrations than those observed at UGR. Aerosol number concentrations at urban environments are strongly linked to local anthropogenic emissions, especially to traffic related emissions. However, aerosol number concentrations at remote sites are rather influenced by long-range transport or/and by transport from lower altitudes in the case of remote mountain sites. Therefore, aerosol

transport from lower altitudes (i.e. from Granada city) could partly explain why aerosol concentrations at SNS were slightly higher than those observed at Puy de Dôme (research station located at 1465 m a.s.l. in central France) or Jungfraujoch (high-alpine station located at 3580 m a.s.l. in the central Swiss Alps), where $N_{10-1000}$ were $2.5 \times 10^3$ cm$^{-3}$ in summer (Venzac et al., 2009) and 767 cm$^{-3}$ in July (Weingartner et al., 1999), respectively.

Table 1 also shows that the ratios between aerosol number concentrations measured at SNS and UGR are similar for all size

ranges ($N_{Tot}$ – 41%, $N_{4-25}$ – 43%, $N_{25-100}$ – 44% and $N_{100-500}$ – 44%). Consequently, the contributions of each aerosol mode to the total aerosol number concentration are similar at both locations. For example, accumulation mode ($N_{100-500}$) has a contribution of 11% at both sites, whereas Aitken mode ($N_{25-100}$) contributed 40% to the total aerosol number concentrations at both measurement sites. Finally, nucleation mode ($N_{4-25}$) is the main contributor to the total aerosol number concentration at both sites during the whole campaign, representing 49% of the total aerosol number concentration at both sites.

*[Table 1]*

Although the contributions of the different modes to total particle number concentration are similar at both sites, Fig. 2 shows that diurnal patterns differ from one site to another, revealing that the aerosol concentrations at both sites are influenced by different sources and mechanisms. $N_{25-100}$ and $N_{100-500}$ diurnal patterns at UGR station show two peaks (one in the morning and other in the evening) in coincidence with those observed in black carbon (BC; a good tracer of traffic emissions; Lyamani et

al., 2011) diurnal pattern (Fig. S1). These diurnal patterns are evidence for the large impact of anthropogenic emissions, mainly traffic emissions, on the aerosol concentrations at UGR site. Similar pattern is observed for $N_{Tot}$ at this site, but with a third peak overlapping the morning traffic rush hours peak. This third peak is observed around 10-12h UTC in coincidence with an increase of nucleation mode aerosol number concentration ($N_{4-25}$). This peak is related to the occurrence of NPF events at UGR due to an increase in atmospheric photochemistry as can be seen later. On the other hand, the total aerosol concentration at

SNS shows an evident diurnal cycle with a peak around 12-16h UTC. This peak is probably associated with the evolution of the boundary layer that transports gases and aerosol from lower altitudes (Fig. S1) or to the occurrence of NPF events at SNS site, since an increase in aerosol nucleation mode aerosol concentration is also observed.



*[Figure 2]*

Although on average the total particle number concentration at UGR is larger than at SNS station, temporal evolution of $N_{Tot}$ shows that during some days the total aerosol concentrations at SNS was higher than those observed at UGR urban station. These high aerosol total concentrations at SNS are also associated with high $N_{4-25}$ concentrations, both peaking around 12h

UTC. The observed peaks in $N_{4-25}$ concentrations (~4 $\times$ 10$^3$ cm$^{-3}$) are more than 40 times larger than the background concentrations (~100 cm$^{-3}$) observed during night-time, suggesting that these cases are probably associated with strong NPF events occurring at SNS site. Due to the observed high contribution of nucleation mode and the suggested relevance of NPF events at both sites, an in-depth study of NPF events is carried out in the next section.

### 3.2 New particle formation analysis

The datasets of aerosol number size distributions at SNS and UGR were analysed and days were classified as NPF events, non-events or undefined and bad-data days following the procedure proposed by Dal Maso et al. (2005). The results are summarised in Table 2. At both sites, a high NPF event frequency (>70%) is observed with 19 and 23 events at SNS and UGR, respectively. For many years, it was thought that NPF events cannot take place in heavily polluted urban areas, since the high condensation sink (high pre-existing aerosol concentration) in these areas was considered detrimental in suppressing

the formation and growth of particles. However, our results show slightly higher frequency of NPF events at UGR urban site in comparison to the remote site, which could be attributed to the higher availability of precursor gases at the urban site. Our results oppose Boulon et al. (2011) observations, who found that nucleation process is more frequent in a high-altitude site than in nearby urban area. Currently, the cause of the high frequency of NPF in polluted urban atmospheres, such as those in Chinese megacities, is still an open question (Kulmala et al., 2017; Yao et al., 2018).

*[Table 2]*

In general, event days at SNS station seem to be connected to the events observed at UGR station, as all the event days observed at SNS are also classified as event days in UGR station. This results in 17 coincident event days at both sites, and suggests a regional scale of NPF events. In this work, class I events have been considered when a growing mode is observed below 25 nm and the retrieval of GR in the 7-25 nm size range is possible. In this sense, at SNS station 10 of the 19 observed events

were classified as class I events, while at UGR station 17 of the 23 observed events were classified as class I events.

It is important to note that the $GR_{7-25}$ retrieval is not unambiguous for some NPF events occurring at SNS station due to the appearance of two modes at same time in the nucleation range (<25 nm) during these cases (Fig. S2). These two modes are not always well separated, making difficult or impossible the retrieval of growth and formation rates for these cases. In some cases, these two modes show similar temporal evolution, but in other cases, completely different behaviour is observed,

suggesting different precursor vapours origin or different formation processes of both aerosol modes. The appearance of these two nucleation modes suggests the advection of aerosol particles probably from lower altitude levels (i.e. from Granada city)



over mountain site, but also the influence of precursor gases advected over SNS station that lead to the occurrence of NPF events at the proximities of SNS station. Also, this phenomenon could be related to convergence of different types of air masses at SNS station. In this sense, the predominant western winds at SNS during NPF events supports the advection of aerosol particles and precursor gases to SNS from the valley (Fig. S3). The appearance of two modes in the nucleation size range were also observed in other mountain sites (e.g., García et al., 2014; Rose et al., 2017) and the cause for their appearance during some NPF events is an open question as remarked in the review of Sellegri et al. (2019). However, despite transport from lower altitudes is expected, NPF events at SNS site are always observed from the smallest measured sizes of the aerosol size distribution (4 nm), implying that NPF takes place in the vicinity of the SNS station at high altitudes.

In UGR urban site, we observed in the $N_{4-25}$ diurnal pattern a frequent occurrence of a local event around 07:00-08:00 UTC coinciding with the morning traffic rush hours followed by another regional event lasting for few hours (Fig. S2), and then another peak coinciding with the evening rush hour (Fig. 2). The first and third peaks could be attributed to the so-called delayed primary particles which are formed in the atmosphere from precursor gases released from hot vehicle exhaust after it dilutes and cools in ambient air (Rönkkö et al., 2017). This aerosol formation does not depend on photochemical reactions and oxidation processes as those from NPF gas-to-particle conversion. The second peak in nucleation mode particle concentrations usually overlaps with the first one and occurs typically around 10:00 UTC (Fig. 2), being associated with the occurrence of regional NPF events at both stations (Fig. 3). In this sense, the NPF events at UGR station appears in the time range from 9:00 UTC to 12:00 UTC, while NPF events are usually observed after 11:00 UTC at SNS station.

*[Figure 3]*

### 3.3 Growth and formation rate

Figure 4a presents the box-whiskers and time series of GRs recorded at SNS and UGR during the campaign. The $GR_{7-25}$ ranges from 4.5 to 9.5 nm h⁻¹ at SNS, with a mean value (± standard deviation) of 6.9 ± 1.7 nm h⁻¹, and from 2.8 to 6.2 nm h⁻¹ at UGR, with a mean value of 4.5 ± 1.0 nm h⁻¹. Also, $GR_{4-7}$ ranges from 3.0 to 5.9 nm h⁻¹ at SNS, with a mean value of 4.1 ± 0.9 nm h⁻¹, and from 2.6 to 5.0 nm h⁻¹ at UGR, with a mean value of 3.6 ± 0.8 nm h⁻¹. These values are in the range of GRs for urban and high-mountain environments reviewed by Nieminen et al. (2018) and Sellegri et al. (2019). The results show that $GR_{7-25}$ and $GR_{4-7}$ mean values at SNS remote station are larger than those observed at UGR station (Fig. 4a) and that $GR_{7-25}$ is always larger at SNS station (comparing same NPF class I event days) (Fig. 4b). It is surprising that the growth rates, especially $GR_{7-25}$, are on average larger at SNS than at UGR since higher precursor vapour concentrations are expected at the urban environment. This fact could have a special importance on cloud formations, since larger GR at SNS mountain station could be translated to larger survival probability of NPF particles to reach CCN sizes, due to shorter time needed for the growth. Another important result is that the relative difference between GRs at SNS and UGR stations is lower for the small size range, being 53% and 14% for the size range of 7-25 and 4-7 nm, respectively. It is important to note that, although it is not a direct





method, the GR at lowest diameter ranges is a good indicator of the quantity of the condensable gases that contribute to the early growth of new particles. Thus, this result suggests that the initial steps of the aerosol formation have almost similar precursors and availability at both sites, resulting in only 14% relative difference of GR at 4-7 nm range. Also, the results suggest higher differences of the available gas precursors for further steps of the aerosol growth, increasing the differences in

GRs from 14% in 4-7 nm to 53% in 7-25 nm range. Also, the increase of GR as a function of the particle size suggests that the growth of the particles is not only due to the same vapours that form particles through nucleation.

However, due to the difference of altitude, the differences in temperature, RH and UV-B between both stations could contribute, at least partly, to this difference in GRs between SNS and UGR. In this sense, lower temperatures at SNS than at UGR (mean temperatures of 14 and 29 ℃ during event days, respectively) can decrease the evaporation rates, enhancing the

effective condensation and thus particle growth at lower temperatures. $H_2SO_4$ is produced from the $SO_2 + OH \rightarrow SO_3$ reaction and $OH$ radicals are produced from water vapour UV absorption. Thus, higher RH and UV-B radiation at SNS (mean values of 44% and 2.6 W m$^{-2}$ at SNS and 21% and 2.3 W m$^{-2}$ at UGR) can increase the $H_2SO_4$-water nucleation and $H_2SO_4$ production, respectively, and thus enhance particle growth, especially in the initial NPF steps. Also, as Boulon et al. (2011) and Manninen et al. (2010) pointed, ion-mediated nucleation could be promoted at higher altitudes compared to low altitudes, and therefore

can contribute to the observed differences.

   *[Figure 4]*

Figure 5 presents the box-whisker and time series of the Js (Formation Rates, $J_4$ and $J_7$), showing $J_7$ mean values of $1.3 \pm 0.8$ and $1.1 \pm 1.2$ cm$^{-3}$ s$^{-1}$ at SNS and UGR, respectively, and $J_4$ mean values of $2.1 \pm 1.3$ and $3.7 \pm 5.2$ cm$^{-3}$ s$^{-1}$, respectively. The values of $J_4$ and $J_7$ show a large variability at both sites, ranging over one order of magnitude. These results differ from GRs

results, in this case, $J_4$ mean value is 43% larger at UGR station, while $J_7$ mean value is 15% larger at SNS station. This result differs from that reported by Nieminen et al. (2018) since $J_7$ is expected to be higher at stations with higher anthropogenic influence. As can be seen in Fig. 5, GR$_{4-7}$ takes slightly larger values at SNS station, and in contrast, $J_4$ shows significantly larger values at UGR station. This difference again points to the decoupling of the mechanisms leading to the initial particle formation and the subsequent growth of the particles. Also, in contrast to the particle formation rate, the particle growth rate

increases as a function of particle size, suggesting the participation of other vapours than sulfuric acid. However, it is worth to mention that the formation rate of 4 and 7 nm particles are not only affected by the new particle formation rate but also by the scavenging of newly formed particles by coagulation into pre-existing particles.

   *[Figure 5]*

In order to quantify the contribution of sulfuric acid to the initial steps of the particle formation, we estimated the growth due

to sulfuric acid to both sizes 4-7 and 7-25 nm, GR$_{4-7}^{SA}$ and GR$_{7-25}^{SA}$, respectively. Figure 6 shows the resulting sulfuric acid contribution to the total experimental growth rates at SNS and UGR. It is clear that sulfuric acid can only explain a small





fraction of the growth rates retrieved in the ranges 4-7 and 7-25 nm at both measurement sites. The ratio $GR^{SA}_{4-7}/GR_{4-7}$ is 9% at both stationss, and the ratios $GR^{SA}_{7-25}/GR_{7-25}$ are 0.8% and 1% at SNS and UGR, respectively. Thus, sulfuric acid explains similar small fraction of the experimental GRs at both sites during the study period. This behaviour is contrary to expectations, as greater sulfuric acid concentrations would be expected at urban environments, and in consequence higher contribution to

the growth rates at urban site. Despite a proxy sulfuric acid concentration was used here, these results strongly suggest a significant contribution of other vapours in this period at both sites. Despite sulfuric acid is traditionally considered as one of the main factors for NPF events to occur, $SO_2$ and sulfuric acid concentrations are lower at SNS when events take place than on non-events days (figure not shown). This is indicative that sulfuric acid concentrations are sufficient for events to take place but not the factor that drives NPF events, especially at SNS mountain site.

*[Figure 6]*

Recent studies have suggested that volatile organic compounds (VOCs), especially extremely low-volatility ones, play vital roles in NPF processes promoting the initial growth of newly formed particles in the atmosphere (e.g., Tröstl et al., 2016). Unfortunately, VOC measurements are not available for the most part of the period analyzed here. However, simultaneous measurements of VOCs were recorded at SNS and UGR during a short intensive period from 15 to 30 June 2017. Surprisingly,

the analysis of these measurements shows that mean VOCs concentrations at SNS remote site was higher than at UGR urban station. Mean total VOCs concentrations of 9.9 and 5.1 $\mu g \cdot m^{-3}$ were observed at SNS and UGR, respectively, during the period 15 to 30 June. Thus, higher concentrations of VOCs at SNS, doubling approximately the UGR concentrations, could explain, at least partly, the higher growth rates observed at SNS as low-volatile vapours produced by photo–oxidation of VOCs could promote particle growth (e.g., Bianchi et al., 2019; Mohr et al., 2019).

**3.4 Condensation Sink and Survival Parameter**

High pre-existing aerosol particle concentrations might also influence NPF events, inhibiting the process by increasing the competition for available condensable gases. In order to investigate the role of pre-existing aerosols, CS has been retrieved for the whole campaign period and averaged each day for 9.00-12.00 UTC time interval. As expected, the average CS obtained at UGR ($6.7 \times 10^{-3}$ $s^{-1}$) was higher than the average value of $2.2 \times 10^{-3}$ $s^{-1}$ obtained at SNS due to higher anthropogenic aerosol

emissions close to UGR urban site. When considering event and non-event days, the CS was higher on event days (2.9 and 6.7 $\times 10^{-3}$ $s^{-1}$ at SNS and UGR, respectively) compared to non-event days (1.8 and $6.6 \times 10^{-3}$ $s^{-1}$ at SNS and UGR, respectively), probably indicating that CS doesn't play a significant role in NPF processes at both sites. The higher CS on event days, that is especially relevant at SNS station, suggests that the sources of nucleating and condensable vapours are closely connected to the sources of particles contributing to CS, and thus CS the inhibiting role in NPF occurrence is not observed. In any case, the

role of CS in NPF processes differs from one high altitude site to other as Sellegri et al. (2019) pointed out. For example, Boulon et al. (2011) at Puy de Dôme station, Venzac et al., (2008) at the Nepal Climate Observatory Pyramid station and Lv





et al. (2018) at Mount Tai found that higher CS inhibits the occurrence of NPF events. However, Boulon et al. (2010) at Jungfraujoch station, Garcia et al. (2014) at Izaña station and Rose et al. (2015) at Chacaltaya station found that higher CS observed in these sites does not inhibits the occurrence of NPF events. Overall, a detailed understanding of the role of CS in NPF events remains an open question and the chemical composition of CS could play an important role on the NPF processes

(Tuovinen, 2019).

Looking on event days, Fig. 7 shows the relationship between CS and $J_7$ and $GR_{7-25}$ at both measurement sites. As can be seen $J_7$ increases with increasing CS at both measurement sites. This result is contrary to the expected one, since concentration of vapours participating in NPF are expected to decrease with increasing CS due to their faster loss rate. However, this result suggests that CS and the concentrations of condensing vapours are connected. This is supported by our previous observation

showing that both CS and $H_2SO_4$ are higher during the NPF event days than during non-event days. On other hand, $GR_{7-25}$ tends to decrease as CS increases, especially at SNS site (Fig. 7). Thus, this result shows that lower CS favours the growth of particles as should be expected. In this sense, as our previous results suggested, this difference in the relationship between CS and $J_7$ and $GR_{7-25}$ points again to the decoupling of the mechanisms leading to the initial particle formation and the subsequent growth of the particles.

*[Figure 7]*

The survival probability of growing clusters links the competition between their growth and scavenging by pre-existing aerosol particles. The key parameter to estimate the clusters survival was defined by Kulmala et al. (2017) as the ratio between the CS and GR (Eq. 5). Our results show that the survival parameter (*P*) ranges from 2 to 10 at SNS with a mean value of $6 \pm 3$ and from 6 to 24 at UGR with a mean value of $13 \pm 6$. At both sites, the values of $P$ are below the threshold value of 50 for NPF

to take place (Kulmala et al., 2017). Larger survival parameter indicates that a smaller percentage of newly formed particles will survive to greater sizes (smaller survival probability). In this sense, lower $P$ values observed at SNS suggests that survival probability of newly formed particles at this site is higher than at UGR urban station. This result indicates that NPF events at SNS could have a significant impact on cloud formations, since due to the higher aerosol survival probability at the SNS mountain station the newly formed particles can reach CCN sizes.

**3.5 Case study: NPF during Saharan dust intrusion**

Mineral dust contributes to a major fraction of the global coarse mode aerosol load, with an emission rate into the atmosphere currently estimated at 1,000–3,000 Tg yr$^{-1}$ from the Earth's surface (Tegen and Schepanski, 2009). Due to its proximity to the African continent, Granada region is frequently affected by Saharan dust intrusions, especially in summer (Mandija et al., 2017; Valenzuela et al., 2015). These African dust intrusions usually transport large mineral dust loads to our study area (e.g.,

Benavent-Oltra et al., 2019; Lyamani et al., 2006). During the analysed period, there was an intrusion of Saharan dust over the region of study from 24 to 27 June, as confirmed by CALIMA warning system (www.calima.ws; Fig. S4). For the confirmation



of desert dust intrusions over different regions in the Iberian Peninsula, CALIMA network uses information derived from different sources, including models, air mass back-trajectory analysis, synoptic meteorological charts, satellite and surface $PM_{10}$ data. As shown in Fig. 8a, there was a significant increase in the coarse particle number concentration ($N_C$, aerodynamic size range 1-20 µm) during 24-27 June, supporting the presence of desert dust over SNS station during these days. According to the $N_C$ analysis it seems that the dust event ended around 10:00 UTC on 27 June, when $N_C$ began to decrease reaching concentrations similar to those observed before the dust intrusion on 23 June. Figure 8b shows the 24-h $PM_{10}$ mass concentration (starting at 07:00 UTC) and the percentage of mineral aerosol on each measurement filter from 07:00 of 23 June to 07:00 of 29 June. It is worth to mention that the last $PM_{10}$ filter measurement was collected during a period of 48-h from 07:00 UTC of 27 June to 07:00 UTC of 29 June. As can be seen, $PM_{10}$ mass concentrations recorded at SNS remote station were higher than 23 µg m$^{-3}$ from 24 June until 27 June and after this period $PM_{10}$ concentration decreases drastically to 6.8 µg m$^{-3}$ on 27-29 June; typical background $PM_{10}$ concentration reported for European mountain sites (e.g., Dinoi et al., 2017). Also, mineral aerosol component contributed more than 65% to the total aerosol mass concentration for each measurement day, reaching highest mineral contribution on filters 3 and 4 (25 and 26 June) with a relative contribution above 80% (Fig. 8b). Also, high concentrations of $Fe_2O_3$ and Ca (good tracers of desert dust aerosol) were observed (Table 3), confirming again the presence and the large impact of Saharan desert dust on aerosol population over our site during 24-27 June.

As discussed before, high pre-existing aerosol particle loadings have been thought to suppress NPF events due to high condensation and coagulation sinks. In this sense, the NPF events are not expected to occur under desert dust episodes, especially at remote sites where the concentrations of condensable vapours are expected to be low. Also, desert dust intrusions with high concentrations of mineral dust particles are expected to reduce the concentrations of condensing vapours and the clustering by limiting solar radiation and hence photochemical oxidation of gaseous precursors, which also reduce the probability of occurrence of NPF events during desert dust events. However, although Saharan dust and NPF events are not expected to occur simultaneously at SNS site, two NPF events classified as class I were observed at SNS on 24 and 26 June during the desert dust intrusion that occurred in the period 24 to 27 June (Fig. 8c). However, no NPF event was observed on dusty day on 25 June although $PM_{10}$ concentration on this day was similar to that registered on 24 and 26 June. These NPF events were previously classified and analysed on section 3.3, retrieving GRs and Js for the possible cases. Below, we will investigate the factors that promote or inhibit the occurrence of NPF events in the presence of mineral dust.

*[Figure 8]*

The evolution of the averaged daily CS (from 9:00 to 12:00 UTC) during these three days shows higher values during the 24 June (5.6 × 10$^{-3}$ s$^{-1}$), decreasing more than 40% and 70% on 25 (3.2 × 10$^{-3}$ s$^{-1}$) and 26 June (1.6 × 10$^{-3}$ s$^{-1}$), respectively. Thus, CS seems not to be a limiting factor for the occurrence of NPF event on 25 June (dusty day). It is important to note that in these cases, the number concentration of coarse mode particles has been considered for CS retrievals. However, the





contributions of coarse particles to the CS during these cases are below 10%, because the contribution of coarse particles in the continuum regime is proportional to the particles diameter ($D_p$) instead of $D_p^2$ (Pirjola et al., 1999).

Figure 9a shows sulfuric acid and VOCs (24h filter sampling starting 07:00 UTC) concentrations and Fig. 9b shows GRs and Js retrievals at SNS site during 23-28 June period. Mean $J_7$ value of 1.72 cm$^{-3}$ s$^{-1}$ was retrieved for the two NPF events observed

on 24 and 26 June dusty days. This $J_7$ value is slightly higher than the mean value observed for the whole campaign (1.35 cm$^{-3}$ s$^{-1}$), suggesting that during these dusty days there were larger production rates of condensing vapours or more favourable conditions for condensation. The opposite behaviour is found for GR$_{7-25}$, since GR$_{7-25}$ value of 6.1 nm h$^{-1}$ was retrieved for the two NPF events observed on 24 and 26 June, which is slightly lower (14% relative difference) than the value observed during the overall campaign (6.9 nm h$^{-1}$). This lower GR values obtained on NPF events on dusty days may be due to the increase of

CS or to its effectiveness during dusty days, which again points to the decoupling of the mechanisms leading to the initial particle formation and the subsequent growth of the particles.

The GR$_{7-25}$ shows similar values on 26 June (6.0 nm h$^{-1}$) and 24 June (6.3 nm h$^{-1}$). This is contrary to the expected from CS and H$_2$SO$_4$ concentrations, since on 26 June the CS mean value was 70% lower than on 24 June and the estimated sulfuric acid concentration on 26 June ($1.3 \times 10^7$ molec cm$^{-3}$) was significantly higher than on 24 June ($4.0 \times 10^7$ molec cm$^{-3}$). As discussed

before, the estimated concentrations of sulfuric acid could only explain less than 10% of the observed GRs. These results evidence that sulfuric acid and CS don't play a relevant role in the particle growth during the NPF events observed during these dusty days. However, the significantly high VOCs concentration observed on 24 June (8.4 µg m$^{-3}$) as compared to 26 June (5.4 µg m$^{-3}$) can explain at least partly the differences observed on GR from 24 to 26 June.

Finally, the results suggest that sulfuric acid is not a limiting factor for the occurrence of NPF event on 25 June dusty day,

since an increase on sulfuric acid concentrations is observed from 24 to 25 June (Fig. 9a). However, as can be seen in Fig. 9a, VOCs concentration showed a decrease by ~67% from 24 to 25 June. The significant reduction of VOCs concentrations on 25 June could be a possible factor limiting the occurrence of NPF on this dusty day. Thus, the results point to VOCs concentrations points as one of the main driving factors controlling the occurrence of NPF event and subsequent particle growth at SNS site during Saharan dust events.

*[Figure 9]*

As commented before, the chemical composition of CS can play a significant role in the occurrence of NPF events. Recent laboratory and observational studies (Dupart et al., 2012; Nie et al., 2014) revealed that TiO$_2$ and Fe$_2$O$_3$ (which are common components of mineral dust) under UV light could enhance the formation of OH and other radicals that favour oxidation and reduction reactions, promoting the occurrence of NPF during dusty conditions. Thus, TiO$_2$ and Fe$_2$O$_3$, acting as catalysts, could

accelerate atmospheric photochemistry repeatedly since they are not consumed in the photo-catalytic reaction. However, since photocatalysis reaction occurs on the surface of the catalysts, this process strongly depends on the available surface of catalysts.





Unfortunately, surface size segregated data of $TiO_2$ and $F_2O_3$ are not available during our measurement campaign. However, $PM_{10}$ chemical analysis shows that reached their highest concentrations on 25 and 26 June, with $TiO_2$ concentrations of 0.24 $\mu g\ m^{-3}$ on both days and $F_2O_3$ concentrations of 1.91 and 1.89 $\mu g\ m^{-3}$ on 25 and 26 June, respectively (Table 3). Thus, the enhanced photochemical activities induced by photo-catalytic reactions as result of the increased $TiO_2$ and $F_2O_3$ concentrations

together with the high concentrations of $H_2SO_4$ and VOCs observed on 26 June, could explain, at least partly, the occurrence of NPF and the observed high formation rates during this dusty day. It is worth noting that the effectiveness of photochemical reactions in promoting the occurrence of NPF depends largely on the available precursor gases. In this sense, although the concentrations of $TiO_2$ and $F_2O_3$ on June 25 were high and comparable to those observed on 26 June, the low availability of precursor gases on this dusty day seems to be the cause of the non-occurrence of NPF event. On other hand, the concentration

of $TiO_2$ and $F_2O_3$ increased from 24 to 26 June, however the concentrations of VOCs decreased by 26% from 24 to 26 June (Fig. 9a). Summarizing, these findings suggest that $TiO_2$ and $Fe_2O_3$ could promote NPF events during dusty conditions, but the availability of VOCs seems to be the main factors controlling the occurrence of NPF events in this area. To improve our understanding in this topic, further investigation accompanied by multiplatform measurement campaigns is needed.

## 4 Summary and conclusions

We investigated the aerosol size distribution at two stations located close to each other (~20 km), but at different altitudes: urban (UGR; 680 m a.s.l.) and high-altitude remote (SNS; 2500 m a.s.l.) site, both in the area of Granada (Spain), and part of AGORA observatory (Andalusian Global ObseRvatory of the Atmosphere). Our results show the important contribution of nucleation mode aerosols to the total particle number concentration at both sites, with a contribution of 47% and 48% at SNS and UGR, respectively. Despite SNS remote site is less influenced by anthropogenic emissions, temporal evolution of aerosol

total concentration shows larger concentrations at SNS site during some days, associated with high concentrations of nucleation mode particles.

New particle formation (NPF) events were studied in detail due to their high frequency of occurrence (>70% at both sites) and their significant contributions to total aerosol number concentrations. Our analysis suggests that NPF events at SNS remote site are associated with the transport of precursor vapours from lower altitudes by orographic buoyant upward flows. NPF

events at SNS site are always observed from the smallest measured sizes of the aerosol size distribution (4 nm), implying that NPF takes place at or in the vicinity of the high altitude SNS station rather than transported from lower altitudes. However, despite the connection between the occurrence of NPF events at the mountain and urban site, different growth rates (GRs) were observed at both stations; the GRs at SNS site were larger than those at UGR site ($GR_{7-25}$ of 6.9 and 4.5 nm $h^{-1}$ and $GR_{4-7}$ of 4.1 and 3.6 nm $h^{-1}$ at SNS and UGR, respectively). This fact could have a special importance on the production of cloud

condensation nuclei (CCN) and therefore on cloud formations which may affect regional/global climate, since larger GR at mountain sites could be translated to larger survival probability of NPF particles to reach CCN sizes, due to shorter time needed

for the growth. The increase in GR differences between SNS and UGR with increasing size (increasing from 14% in the size range of 4-7 nm to 53% in the range 7-25 nm) suggests higher concentrations of the available gas precursors, other than sulfuric acid, for further steps of the aerosol growth. This was also supported by the low contribution of sulfuric acid to the observed GRs at both sites (<1% and <10% for 7-25 and 4-7 nm size range, respectively). A deep analysis of the role of precursor gases
in NPF processes points to the volatile organic compounds (VOCs) as one of the main factors controlling the NPF events at both sites.

Higher condensation sink (CS) values on event days compared to non-event days were observed at both measurement sites, suggesting that CS is not the limiting factor of NPF processes at either of the sites. In fact, the obtained results suggest that the availability of condensable vapours at SNS is closely connected to CS. In this sense, although Saharan dust and NPF events
are not expected to occur simultaneously at SNS remote site, two NPF events were identified during an intrusion of Saharan dust. A close analysis of these two NPF events suggest that $TiO_2$ and $Fe_2O_3$ could promote NPF events, but the availability of VOCs seems to be the main factors controlling the occurrence of NPF events during dusty conditions. Despite further investigation is needed to improve our understanding in this topic, this result suggests that climate effects of mineral dust and NPF are not disconnected from each other as it was commonly thought. Therefore, since mineral dust contributes to a major
fraction of the global aerosol mass load, dust-NPF interaction should be taken into account in global aerosol-climate modelling for better climate change prediction.

## Data availability

The data used in the manuscript is available from the first author at casquero@ugr.es.

## Author contributions

JACV analysed the data and wrote the manuscript. JACV and HL operated and processed the in-situ measurement and RR operated and processed the UVB measurements. LD, SH and PP helped on the pre-processed of the raw data and provided feedback on the NPF analysis and the MATLAB code. LD, SH, PP and TP provided useful discussion and ideas. The formal analysis, investigation, writing of the original draft, preparation, review of the writing, and editing were performed by JACV, HL and LAA. The project administration and funding acquisition were done by FJOR and LAA. All authors provided
comments on the manuscript and helped with paper correction.

## Competing interests

The authors declare that they have no conflict of interest.



## Acknowledgments

Juan Andrés Casquero-Vera is funded by MINECO under predoctoral program FPI (BES-2017-080015). This work was supported by the Spanish Ministry of Economy and Competitiveness through projects CGL2016-81092-R, CGL2017-90884-REDT and RTI2018-097864-B-I00, by the Andalusia Regional Government through project P18-RT-3820, by the European

Union's Horizon 2020 research and innovation program through project ACTRIS-2 (grant agreement No 654109), ACTRIS-IMP (grant agreement No 871115) and through SMart URBan Solutions for air quality, disasters and city growth (grant agreement No 689443) ERA-NET-Cofund, by the Academy of Finland (311932; 307537; ACTRIS-Finland, 328616; ACTRIS-CF, 329274) and by University of Helsinki (ACTRIS-HY). Simo Hakala acknowledges the doctoral programme in atmospheric sciences (ATM-DP, University of Helsinki) for financial support. The authors thankfully acknowledge the

FEDER programme for the instrumentation used in this work and the University of Granada, which supported this study through the excellence units programme. The authors would like to thank Air Quality Service from Junta de Andalucía (Consejería de Medio Ambiente y Ordenación del Territorio), University of Évora, Fundación CEAM and TSI company for their support during the SLOPE II campaign.

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



**Tables and figures**

**Table 1. Statistical overview of daily aerosol concentrations in different size ranges in $10^3$ cm$^{-3}$ at SNS and UGR station.**

|  | SNS | | | | UGR | | | |
|---|---|---|---|---|---|---|---|---|
|  | $N_{4-25}$ | $N_{25-100}$ | $N_{100-500}$ | $N_{Tot}$ | $N_{4-25}$ | $N_{25-100}$ | $N_{100-500}$ | $N_{Tot}$ |
| Min | 0.2 | 0.4 | 0.1 | 1.0 | 1.1 | 1.2 | 0.4 | 2.4 |
| Median | 1.3 | 1.2 | 0.4 | 3.2 | 3.8 | 3.3 | 1.0 | 9.1 |
| Max | 6.2 | 3.8 | 0.7 | 9.6 | 8.0 | 5.3 | 1.7 | 14.9 |
| Mean | 1.7 | 1.4 | 0.4 | 3.5 | 4.0 | 3.2 | 0.9 | 8.1 |
| St. Dev. | 1.4 | 0.9 | 0.2 | 2.3 | 1.8 | 1.2 | 0.4 | 3.2 |

**Table 2. Number of NPF event (E), non-event (NE) or undefined (UN) NPF days and bad-data (BD) observed at SNS and UGR.**

| Site | E (days) | NE (days) | UN (days) | BD (days) |
|---|---|---|---|---|
| SNS | 19 | 5 | 3 | 5 |
| UGR | 23 | 4 | 1 | 4 |

**Table 3. Concentration of Ca, K, Fe₂O₃ and Fe₂O₃ during the period 23-29 June.**

| (µg m⁻³) | Filter 1 23 June | Filter 2 24 June | Filter 3 25 June | Filter 4 26 June | Filter 5 27-29 June |
|---|---|---|---|---|---|
| Ca | 1.05 | 0.91 | 1.22 | 0.94 | 0.46 |
| K | 0.30 | 0.34 | 0.53 | 0.47 | 0.13 |
| Fe₂O₃ | 1.00 | 1.18 | 1.91 | 1.89 | 0.33 |
| TiO₂ | 0.13 | 0.15 | 0.24 | 0.24 | 0.04 |


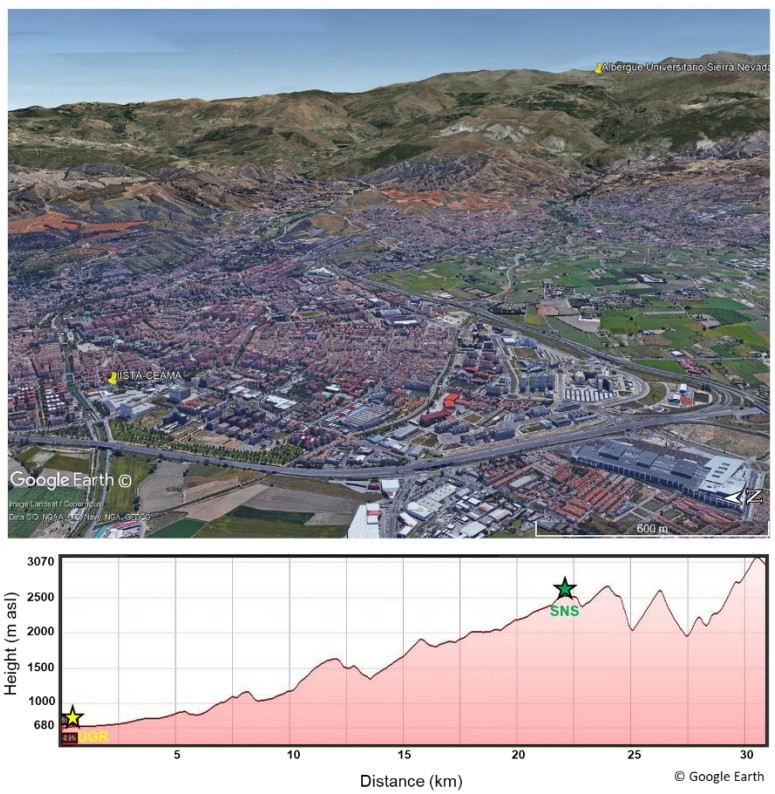

**Figure 1. Map of Granada and Sierra Nevada and topographic profile between UGR and SNS stations. Yellow pins show the location of UGR and SNS stations.**

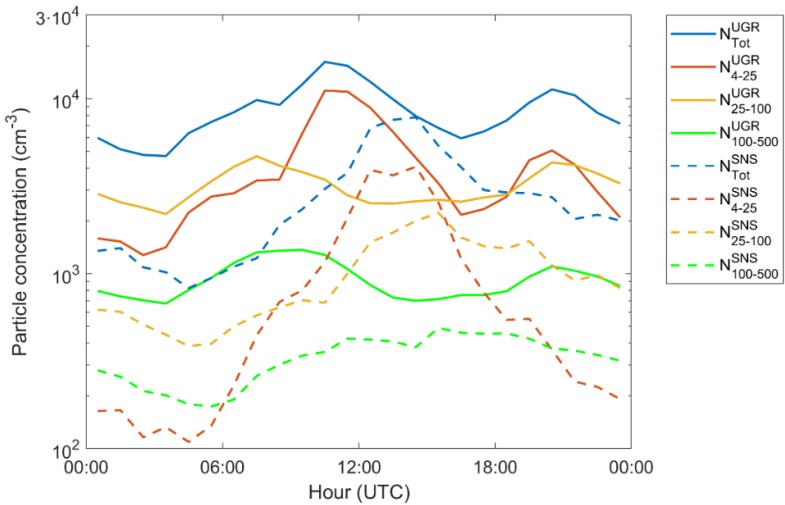

5      **Figure 2. The mean diurnal variation of particle concentrations in different size ranges at SNS and UGR for the whole analysed period.**





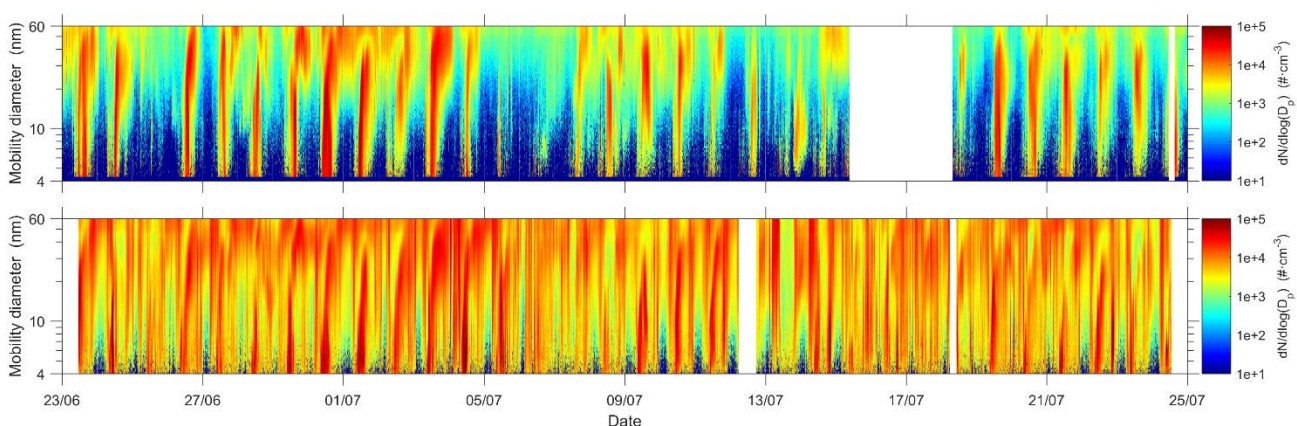

**Figure 3. Size distribution time evolution from 23 June to 24 July at SNS (top) and UGR (bottom) station.**

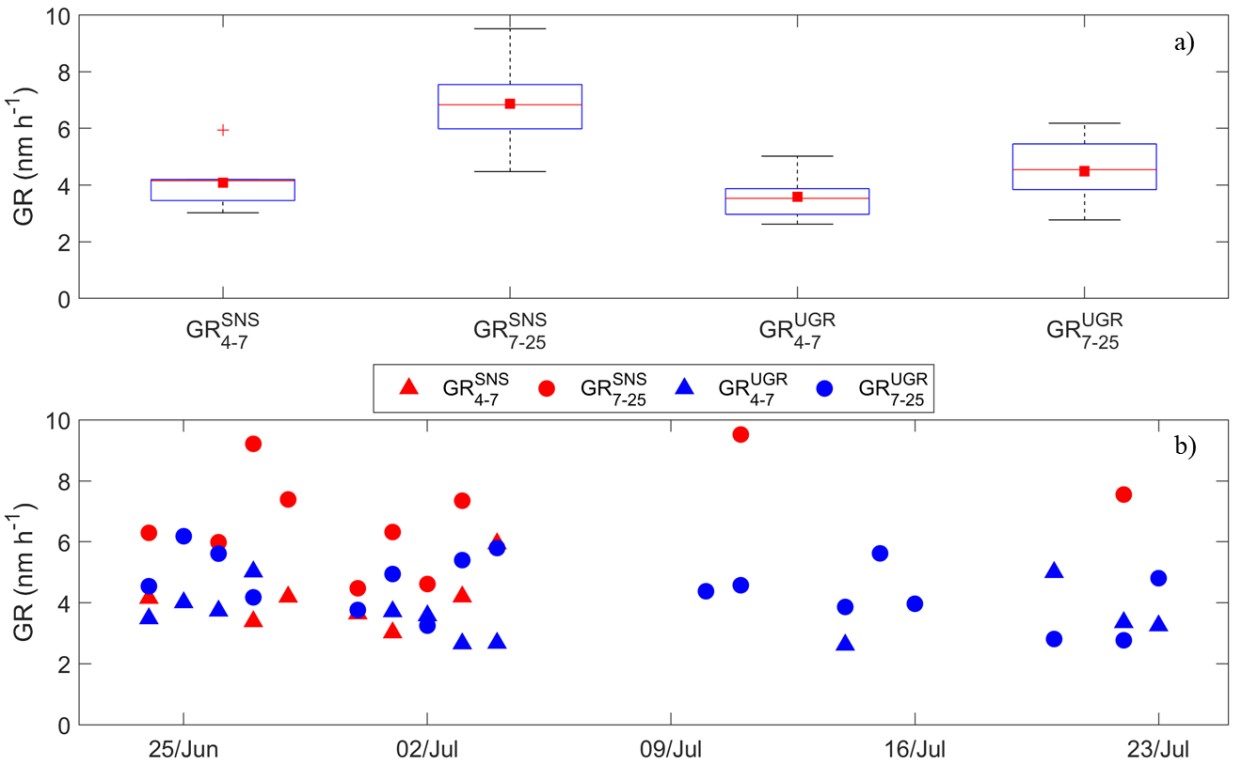

**Figure 4. (a) Box and whisker plot of growth rates recorded at SNS and UGR stations. The red line represents the median of the data, the square represents the mean of the data and the lower and upper edges of the box represent 25th and 75th percentiles of the data, respectively. The length of the whiskers represents 1.5× interquartile range which includes 99.3% of the data. (b) Growth rates time series at SNS and UGR station.**





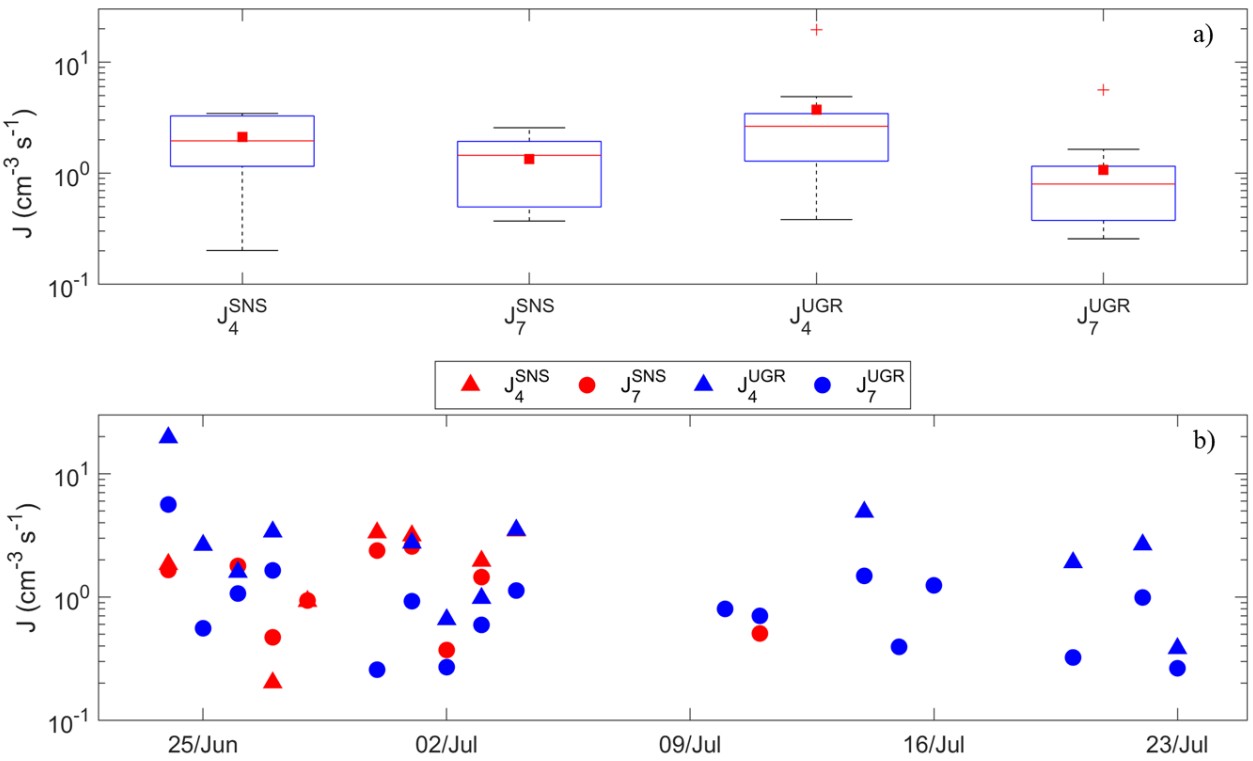

**Figure 5. (a) Box and whisker plot of formation rates recorded at SNS and UGR stations. The red line represents the median of the data, the square represents the mean of the data and the lower and upper edges of the box represent 25th and 75th percentiles of the data, respectively. The length of the whiskers represents 1.5× interquartile range which includes 99.3% of the data. (b) Formation rates time series at SNS and UGR station.**

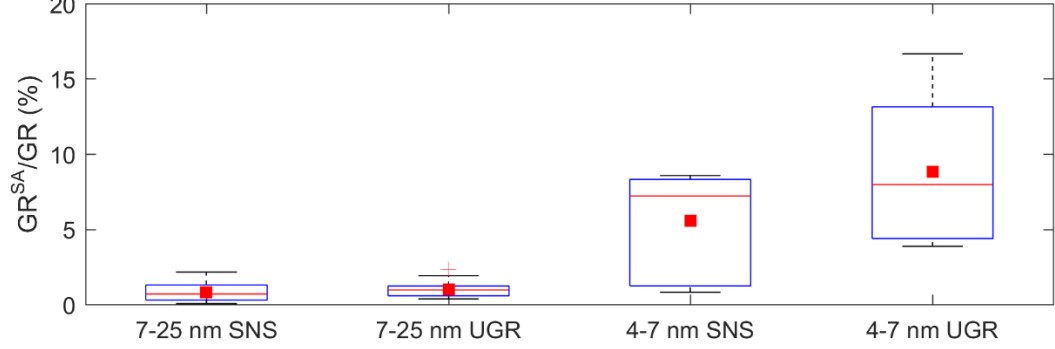

**Figure 6. Box and whisker plot of sulfuric acid contribution to the total measured growth rate at SNS and UGR. The red line represents the median of the data, the square represents the mean of the data and the lower and upper edges of the box represent 25th and 75th percentiles of the data, respectively. The length of the whiskers represents 1.5× interquartile range which includes 99.3% of the data.**





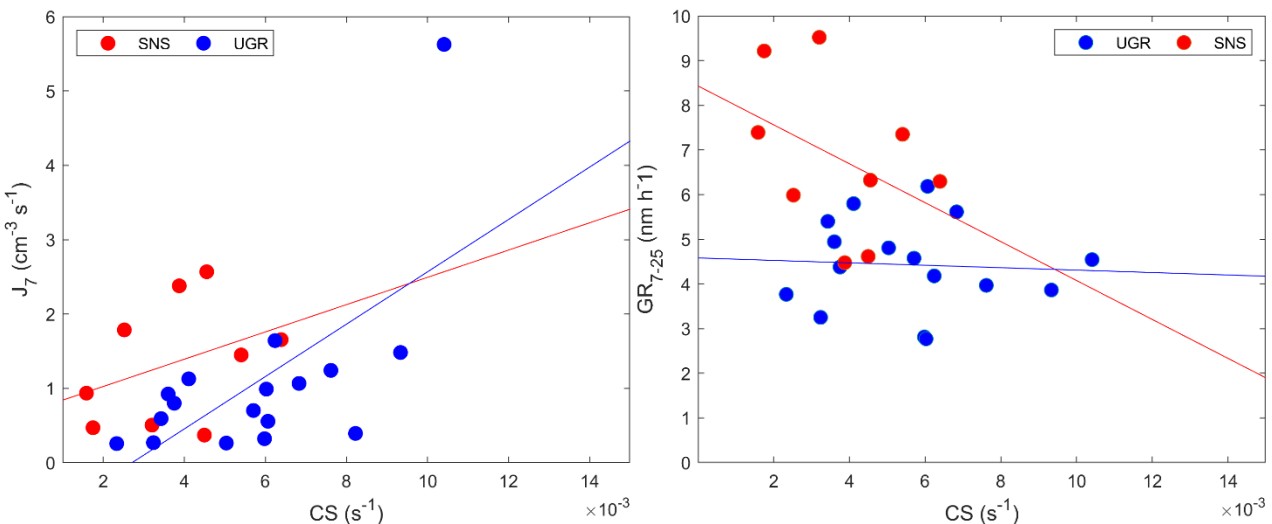

**Figure 7. Relationship between CS (time window: 9:00-16:00 UTC) and J7 (left) and GR7-25 (right) during study period.**

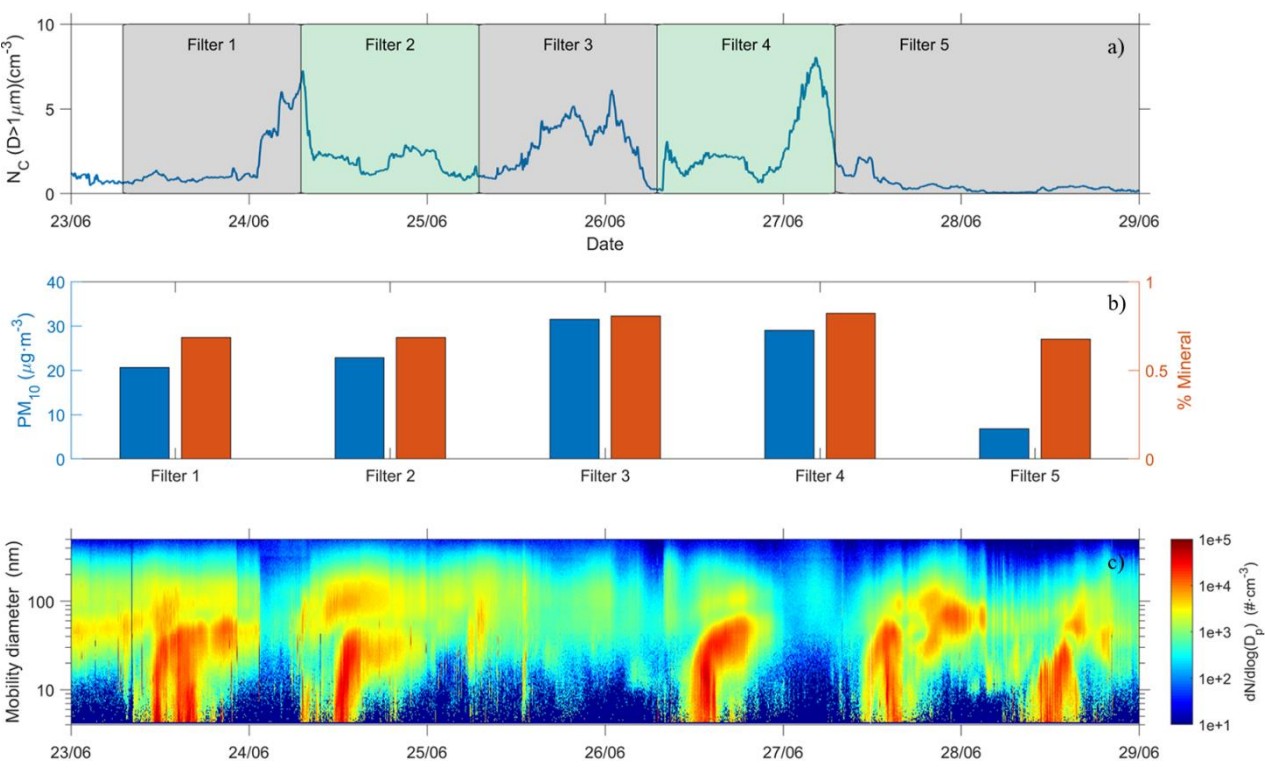

5   **Figure 8. Temporal evolution of (a) coarse mode aerosol number concentration, (b) PM$_{10}$ concentration and mineral matter fraction (%) and (c) aerosol size distribution from 23 to 29 June at SNS station.**





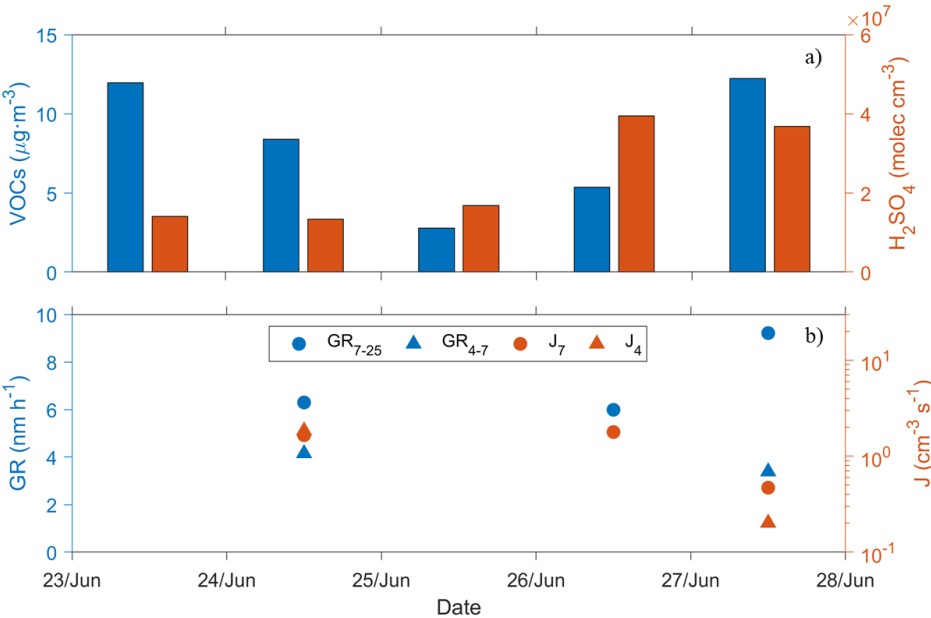

Figure 9. (a) VOCs and H₂SO₄ concentration and (b) GR and J in different size ranges from 23 to 28 June observed at SNS station.