# Peer review of "New particle formation at urban and high-altitude remote sites in the south-eastern Iberian Peninsula"

_Atmospheric Chemistry and Physics, 2020_

## Referee Comment (RC1) · Anonymous Referee #1 · 25 Jun 2020

This manuscript attempts to study NPF events at two contrastive sites, i.e., urban and high-altitude remote sites within 20 km distance. The authors found that NPF was associated with the transport of gaseous precursors from lower altitudes, always observed from the smallest measured sizes and had a higher growth rate of newly formed particles at the high-altitude site. They also analyzed the contribution of sulfuric acid in particle growth, the importance of CS and availability of VOC in NPF events. In my opinion, the paper is generally well-written and suitable for publishing in ACP. A few minor comments are listed for the authors considering.

1) Page 3, lines 3-10, the authors are encouraged to cite the papers, in which the

contributions of NPF to CCN at specific supersaturations were measured directly. 2) Page 7, the bottom paragraph and Page 8, the top paragraph, the reviewer gets lost by the comparison presented here. The similarity and difference between the contrastive sites in literature should be compared with the findings in this study, correct? 3) Page 9, lines13-15, "For many years, it was thought that NPF events cannot take place in heavily polluted urban areas, since the high condensation sink (high pre-existing aerosol concentration) in these areas was considered detrimental in suppressing the formation and growth of particles', What situation is "heavily polluted", please add a quantitative definition. 4) Page 10, lines 28-29, "This fact could have a special importance on cloud formations, since larger GR at SNS mountain station could be translated to larger survival probability of NPF particles to reach CCN sizes, due to shorter time needed for the growth." This is not necessarily true by considering the ceiling of particle growth from 10 nm to CCN size or even particle shrinkage, e.g., Man et al., EST, ‏ 49, ‏ 7170-7178,‏ JUN 16 2015. 5) Section 3.3, the aerosol acidity and aerosol phase state may also affect the growth rate of newly formed particles, please add the analysis if possible. 6) Page 13, lines 15-24, the reviewer has the same concern as presented comment 3 7) Section 3.5, no direct measurements of sulfuric acid vapor are one of major weaknesses here, and the weakness should be added.

---

## Referee Comment (RC2) · Anonymous Referee #2 · 4 Sep 2020

The MS mainly deals with investigating the aerosol size distributions in the diameter range of 4-500 nm measured simultaneously at two close locations with rather different altitudes in southern Spain during an intensive summer campaign to study the occurrence frequency and other characteristics of NPF events. It is a very complex and extended study. It is of interest for international research community, and I suggest that it is published in the ACP. There are, however, several issues to be improved or corrected.

1. A general comment. This MS is not easy to read since it contains a large variety of details, which are sometimes not well organized. The authors may want to improve

several of its parts (e.g. P15, L26–P16, L14).

2. P12, L3–9 and L26–30: The discussions the roles and importance of H2SO4 and partially of CS in the formation process at the sites seem to be speculative. The authors should provide further firm evidence and additional explanations of their ideas, or they should reduce and simplify this part.

3. P5, L26–28 and later P6, L15–17: The association of the three modes in the size distributions to size fractions should be proved by showing a typical size distribution and the size ranges. Does the position of the modes change in time at the sites? The authors may want to explain why they did not use modal areas from fitting, which was performed anyway, instead of size fractions.

4. P5, L14–16: Further details on the "complete chemical analysis" should be given. What are the species measured, what are the analytical methods used and their performance properties?

5. P6, L15–19: It is not completely clear what size interval do CoagSDp and NDp represent, and how they were calculated?

6. P8, L18–19: How can you explain the high contribution of the nucleation mode (49%) to the total particle numbers on non-NPF even days?

Minor comments Abstract and P3, L30 and L32: Abbreviation a.s.l. should be resolved. P5, L6: Abbreviations RH and PSL should be resolved. P8, L32: Rewording is required (in aerosol nucleation mode aerosol concentration). P9, L10–11: Repetition; it should be removed.

---

## Author Response (AR1)

**Review of Casquero-Vera et al. "New particle formation at urban and high-altitude remote sites in the south-eastern Iberian Peninsula" by Anonymous Referee #1**

5 **We thank the reviewer for his/her valuable comments and suggestions that helped us to improve the quality of the manuscript. Our responses to the reviewer's comments are detailed below. Our answers to reviewer are shown in bold and the changes inserted in the manuscript are noted here in italic and between quotation marks. The changes in the new version of the manuscript are noted in red.**

**Response to referee #1:**

10 This manuscript attempts to study NPF events at two contrastive sites, i.e., urban and high-altitude remote sites within 20 km distance. The authors found that NPF was associated with the transport of gaseous precursors from lower altitudes, always observed from the smallest measured sizes and had a higher growth rate of newly formed particles at the high-altitude site. They also analyzed the contribution of sulfuric acid in particle growth, the importance of CS and availability of VOC in NPF events. In my opinion, the paper is generally well-written and suitable for publishing in ACP. A few minor comments are listed

15 for the authors considering.

**A point by point response is included below.**

1) Page 3, lines 3-10, the authors are encouraged to cite the papers, in which the contributions of NPF to CCN at specific supersaturations were measured directly.

20 **Following the reviewer's suggestions, we included papers in which the contribution of NPF to CCN were measured directly and made the following changes:**

*"Typically, the size at which aerosols activate as CCN ranges from 50 to 150 nm (Kerminen et al., 2012). However, some observational studies have showed that particles do not have to grow to sizes above 50 nm to be able to act as CCN and particles as small as 20 nm can form cloud droplets (e.g., Fan et al., 2018; Leaitch et al., 2016; Leng et al., 2014). In this*

25 *sense, NPF events are one of the main processes producing aerosols in these sizes that has been estimated to enhance CCN number concentration by a factor of 1.2-1.8, depending on supersaturation (e.g., Dameto de España et al., 2017; Leng et al., 2014)."*

2) Page 7, the bottom paragraph and Page 8, the top paragraph, the reviewer gets lost by the comparison presented here. The similarity and difference between the contrastive sites in literature should be compared with the findings in this study, correct?

**We agree with the reviewer that the comparison presented in the first paragraph of section 3.1 was not clear and different size ranges were used in this comparison. In order to improve this section, we made the following changes in the new manuscript version:**

*"In general, the direct comparison of the results obtained in this study with those reported in literature is difficult and large differences in the aerosol number concentrations between the different sites may result from differences in the measured size ranges, instrumentation, sampling period, site location and proximity to the sources. However, the average aerosol concentration measured at UGR urban station was in the range of those obtained in summer season ($4\text{-}24 \times 10^3$ cm$^{-3}$) in other European urban sites (e.g., Birmili et al., 2016; Gómez-Moreno et al., 2011; Pérez et al., 2010; Pey et al., 2010). On the other hand, the mean aerosol concentration at SNS was slightly higher than that observed at Puy de Dôme (research station located at 1465 m a.s.l. in central France), where $N_{10-1000}$ were $2.5 \times 10^3$ cm$^{-3}$ in summer (Venzac et al., 2009). However, the mean aerosol concentrations at SNS was significantly higher than that reported (mean $N_{10-1000}$ of 767 cm$^{-3}$ in July) for Jungfraujoch (high-alpine station located at 3580 m a.s.l. in the central Swiss Alps) by Weingartner et al. (1999). Aerosol number concentrations at remote sites are rather influenced by long-range transport or/and by transport from lower altitudes in the case of remote mountain sites. Therefore, aerosol transport from lower altitudes (i.e. from Granada city) could partly explain the high aerosol concentration observed over SNS station."*

3) Page 9, lines13-15, "For many years, it was thought that NPF events cannot take place in heavily polluted urban areas, since the high condensation sink (high pre-existing aerosol concentration) in these areas was considered detrimental in suppressing the formation and growth of particles', What situation is "heavily polluted", please add a quantitative definition.

**We agree with the reviewer that heavily polluted is not a quantitative definition and we removed this term along the manuscript. We made the following changes in the new manuscript version:**

*"For many years, it was thought that NPF events cannot take place in urban areas, where the high pre-existing aerosol concentrations was considered detrimental in suppressing the formation and growth of particles due to high condensation sink."*

4) Page 10, lines 28-29, "This fact could have a special importance on cloud formations, since larger GR at SNS mountain station could be translated to larger survival probability of NPF particles to reach CCN sizes, due to shorter time needed for

the growth." This is not necessarily true by considering the ceiling of particle growth from 10 nm to CCN size or even particle shrinkage, e.g., Man et al., EST, âAˇR 49, â´AˇR´ 7170-7178,âAˇR JUN 16 2015.

**We agree with the reviewer that larger GR does not necessarily mean that NPF particles will reach CCN sizes and that ceiling or shrinkage should be considered as limiting factors. In our case, ceiling size varies from 25 to 60 nm during class I events at SNS site and no shrinkage process was observed. That means that NPF events at SNS site will affect to CCN concentrations at least at high SS. Thus, we included the considerations made by the reviewer as follow.**

*"This fact could have a special importance on cloud formations, since larger GR at SNS mountain station could be translated to larger survival probability of NPF particles to reach CCN sizes, due to shorter time needed for the growth. However, this fact is of importance after considering that ceiling size and shrinkage are not limiting factors to reach CCN size. Since ceiling of particle growth mode ranges from 25 to 60 nm with a mean value of 41 nm during class I events at SNS site and shrinkage has not been observed at this site, NPF events at SNS site can affect CCN concentrations at least at high SS (e.g., Fan et al., 2018; Leaitch et al., 2016; Leng et al., 2014)."*

5) Section 3.3, the aerosol acidity and aerosol phase state may also affect the growth rate of newly formed particles, please add the analysis if possible.

**We agree with the reviewer that aerosol acidity and aerosol phase state affect the growth rate of newly formed particles but we can not estimate the evaluate of these mechanisms in our sites with the current available measurements. In this sense, we included in section 3.3 these mechanisms as possible processes involved on the observed differences.**

*"Also, as Boulon et al. (2011) and Manninen et al. (2010) pointed, ion-mediated nucleation could be promoted at higher altitudes compared to low altitudes, and therefore can contribute to the observed differences. However, as Lehtipalo et al. (2016) showed, this mechanism only accelerates the growth of newly formed particles at low concentrations of base compounds. When a strongly basic compound is present, the growth of newly formed particles can be greatly enhanced by acid-base clusters. Thus, the presence of stabilizing vapours could also be responsible of the differences observed."*

6) Page 13, lines 15-24, the reviewer has the same concern as presented comment 3.

**To avoid confusion, we removed lines 7-9 of P14 in the new manuscript version.**

7) Section 3.5, no direct measurements of sulfuric acid vapor are one of major weaknesses here, and the weakness should be added.

**We agree no direct measurements of sulfuric acid is one weakness in this work. In section 2.2 (Data analysis section) of ACPD version we already mentioned that: "***The choice of k value will have an influence on the absolute value of the*** $H_2SO_4$ ***concentrations, but not on the relative variability***", and we also made the following changes in the new version of Results section (P15, lines 27-31 on new manuscript version):**

*"The $GR_{7-25}$ shows similar values on 26 June (6.0 nm h$^{-1}$) and 24 June (6.3 nm h$^{-1}$). This is contrary to the expected from CS and $H_2SO_4$ concentrations. Firstly, on 26 June the CS mean value was 70% lower than on 24 June. Secondly, despite no direct measurements of sulfuric acid were available, the sulfuric acid concentration estimated by proxy on 26 June (4.0 × 10$^7$ molec cm$^{-3}$) was 3 times higher than on 24 June (1.3 × 10$^7$ molec cm$^{-3}$). As discussed before, the estimated concentrations of sulfuric acid could only explain less than 10% of the observed GRs."*

**References**

- Leaitch, et al., 2016: Effects of 20–100 nm particles on liquid clouds in the clean summertime Arctic.

- Leng et al., 2014: Impacts of new particle formation on aerosol cloud condensation nuclei (CCN) activity in Shanghai: case study.

- Fan et al., 2018: Substantial convection and precipitation enhancements by ultrafine aerosol particles.

- Dameto de España et al., 2017: Long-term quantitative field study of New Particle Formation (NPF)events as a source of Cloud Condensation Nuclei (CCN) in the urban background of Vienna.

- Lehtipalo et al. 2016: The effect of acid–base clustering and ions on the growth of atmospheric nano-particles.

**Review of Casquero-Vera et al. "New particle formation at urban and high-altitude remote sites in the south-eastern Iberian Peninsula" by Anonymous Referee #2**

5 **We would like to acknowledge the work done by the referee in the revision of our manuscript. We appreciate his/her effort and contributions to improve the quality of the paper. Our responses to the reviewer's comments are detailed below. Our answers to reviewer are shown in bold and the changes inserted in the manuscript are noted here in italic and between quotation marks. The changes in the new version of the manuscript are noted in red.**

**Response to referee #2:**

10 The MS mainly deals with investigating the aerosol size distributions in the diameter range of 4-500 nm measured simultaneously at two close locations with rather different altitudes in southern Spain during an intensive summer campaign to study the occurrence frequency and other characteristics of NPF events. It is a very complex and extended study. It is of interest for international research community, and I suggest that it is published in the ACP. There are, however, several issues to be improved or corrected.

15 **A point by point response is included below.**

1) A general comment. This MS is not easy to read since it contains a large variety of details, which are sometimes not well organized. The authors may want to improve several of its parts (e.g. P15, L26–P16, L14).

**Following reviewers' suggestions, we have rewritten the paragraph P16, L10–25 as follows:**

20 *"Finally, the chemical composition of desert dust particles (acting as CS) may be another important factor that can play a significant role in the occurrence of NPF events during dust events. In fact, recent laboratory and observational studies (Dupart et al., 2012; Nie et al., 2014) revealed that the presence of $TiO_2$ and $Fe_2O_3$ (which are common components of mineral dust) under UV light could promote the occurrence of NPF can enhance the formation of OH and other radicals that favour oxidation reactions, promoting the occurrence of NPF during dusty conditions. These components, acting as catalysts, are not*

25 *consumed in the photo-catalytic reaction and can accelerate atmospheric photochemistry repeatedly. The $PM_{10}$ elemental analysis shows that both components reached their highest concentrations on 25 and 26 June, with $TiO_2$ concentrations of 0.24 µg $m^{-3}$ on both days and $F_2O_3$ concentrations of 1.91 and 1.89 µg $m^{-3}$ on 25 and 26 June, respectively (Table 3). It is worth noting that the effectiveness of photochemical reactions in promoting the occurrence of NPF depends largely on the available precursor gases. Thus, the increased $TiO_2$ and $F_2O_3$ concentrations together with the high concentrations of $H_2SO_4$*

*and VOCs observed on 26 June, could explain, at least partly, the occurrence of NPF and the observed high formation rates during this dusty day (26 June). However, although the concentrations of $TiO_2$ and $F_2O_3$ on June 25 (non-event day) were high and comparable to those observed on 26 June, the low availability of precursor gases on this dusty day seems to be the cause of the non-occurrence of NPF event. Summarizing, these findings suggest that $TiO_2$ and $Fe_2O_3$ could promote NPF events during dusty conditions, but the availability of VOCs seems to be the main factors controlling the occurrence of NPF events in this area. To improve our understanding in this topic, further investigation accompanied by multiplatform measurement campaigns is needed."*

**And in the last paragraph of section 3.2 we made the following change:**

*"While regional NPF events are usually observed after 11:00 UTC at SNS station probably due to the upslope transport of precursors vapours from lower altitudes, at UGR urban site, we observed the frequent occurrence of a local event (burst event) around 07:00-08:00 UTC followed by another event lasting for few hours (see Fig. S2). First event at urban site coincides with morning traffic rush hours and could be attributed to the so-called delayed primary particles which are formed in the atmosphere from precursor gases released from hot vehicle exhaust after it dilutes and cools in ambient air (Rönkkö et al., 2017). This local event is followed by another event that is associated with the occurrence of regional NPF events at both stations (Fig. 3). The regional NPF events at UGR station appear in the time range from 9:00 UTC to 12:00 UTC, while NPF events at SNS station are usually observed after 11:00 UTC. It is worth to mention that the production of 4 nm particles can be observed at SNS even though their production has already ended at UGR, which suggests that completely new particles are being formed at high altitudes."*

2) P12, L3–9 and L26–30: The discussions the roles and importance of H2SO4 and partially of CS in the formation process at the sites seem to be speculative. The authors should provide further firm evidence and additional explanations of their ideas, or they should reduce and simplify this part.

**We have reduced both paragraphs as follows:**

*"In order to quantify the contribution of sulfuric acid to the initial steps of the particle formation, we estimated the growth due to sulfuric acid in both 4-7 and 7-25 nm size ranges, $GR^{SA}_{4-7}$ and $GR^{SA}_{7-25}$, respectively. Figure 6 shows the resulting sulfuric acid contribution to the total experimental growth rates at SNS and UGR. It is clear that sulfuric acid can only explain a small fraction of the growth rates retrieved in the ranges 4-7 and 7-25 nm at both measurement sites. The ratio $GR^{SA}_{4-7}/GR_{4-7}$ is 9% at both stations, and the ratios $GR^{SA}_{7-25}/GR_{7-25}$ are 0.8% and 1% at SNS and UGR, respectively. Thus, sulfuric acid explains similar small fraction of the experimental GRs at both sites during the study period and, despite a proxy sulfuric acid concentration was used here, these results strongly suggest a significant contribution of other vapours in this period at both sites. Furthermore, despite sulfuric acid is traditionally considered as one of the main factors for NPF events to occur, $SO_2$*

*and sulfuric acid concentrations are lower at SNS when events take place than on non-events days (figure not shown), indicating that sulfuric acid concentrations are sufficient for events to take place but not the factor that drives NPF events."*

*"When considering event and non-event days, the CS was higher on event days (2.9 and 6.7 × 10-3 s-1 at SNS and UGR,* respectively) compared to non-event days (1.8 and 6.6 × 10-3 s-1 at SNS and UGR, respectively), indicating that CS does not play a significant role in NPF processes at both sites. It is worth mentioning that the role of CS in NPF processes differs from one high altitude site to other as Sellegri et al. (2019) pointed out. For example, Boulon et al. (2011) at Puy de Dôme station, Venzac et al., (2008) at the Nepal Climate Observatory Pyramid station and Lv et al. (2018) at Mount Tai found that higher CS inhibits the occurrence of NPF events. However, Boulon et al. (2010) at Jungfraujoch station, Garcia et al. (2014) at Izaña station and Rose et al. (2015) at Chacaltaya station found that higher CS observed in these sites does not inhibits the occurrence of NPF events. Overall, a detailed understanding of the role of CS in NPF events remains an open question and the chemical composition of CS could play an important role on the NPF processes (Tuovinen et al., 2020)."*

3) P5, L26–28 and later P6, L15–17: The association of the three modes in the size distributions to size fractions should be proved by showing a typical size distribution and the size ranges. Does the position of the modes change in time at the sites? The authors may want to explain why they did not use modal areas from fitting, which was performed anyway, instead of size fractions.

**In order to prove the association of three modes in the size distributions, Figure R1 shows the mean size aerosol distribution for the days presented in Fig. S2 of Supplementary Material. Figure S2 and Fig. R1 show that nucleation and Aitken mode are well separated at both sites with tinny contribution of accumulation mode as it is pointed in section 3.1 of ACPD manuscript. We agree that the use of modal areas is a useful information that gives information of aerosol dynamics and the processes involved on it. However, dynamics of aerosol size distribution has been studied in following sections using other parameters as growth rates. In addition, since the position of the modes will change on time, the use of modal areas from fitting makes impossible the comparison of aerosol concentrations at both sites.**

[Figure]

**Figure R1. Example of diary mean aerosol size distribution at UGR and SNS sites.**

The use of the aerosol size ranges presented in this work are commonly used to provide information about sources and atmospheric processing of particles (Hama et al., 2017 and references therein). In this sense, the nucleation mode (particle diameter < 25 nm) can originate from primary combustion particles emitted directly into the atmosphere and new particles formed in the atmosphere by gas-to-particle conversion. Aitken mode (25 nm < particle diameter < 100 nm) may include a mixture of soot particles emitted in combustion processes and coagulated nucleation mode particles. Accumulation mode particles (100 nm < particle diameter < 1000 nm) may be result from biomass and fuel combustion processes and coagulation and growth of Aitken mode aerosols by condensation of vapors onto existing particles. Therefore, the analysis of the mode-segregated particle concentrations can provide some information upon the sources and processes contributing to particle number concentrations.

In light of the above, we have included the following sentences in P6, L2–7 of new manuscript version:

*"In order to gain some insight upon the sources and processes contributing to particle number concentrations over both sites, we segregated the 5-min particle size distributions measured by SMPS into three diameter ranges: nucleation mode from 4 to 25 nm ($N_{4-25}$), Aitken mode from 25 to 100 nm ($N_{25-100}$) and accumulation mode from 100 to 500 nm ($N_{100-500}$). This is because these distinct particle modes result from different emission sources and chemical and physical processes (e.g., Hama et al., 2017 and reference therein). In addition, total particle number concentration is calculated from the whole 4 to 500 nm range ($N_{Tot}$)."*

4) P5, L14–16: Further details on the "complete chemical analysis" should be given. What are the species measured, what are the analytical methods used and their performance properties?

**We have made the following changes in the new manuscript version (P5, L17-21):**

*"The filters were conditioned and treated pre- and post-sampling and major elements (Al, Ca, K, Mg, Fe, Ti, Mn, P, Na) were determined by inductively coupled plasma mass spectrometry (ICP-MS) at Institute of Environmental Assessment and Water Research (IDAEA-CSIC, Barcelona, Spain) following the procedure of Querol et al. (2001). Detection limit and accuracy were estimated in 0.02 ng m⁻³ and 3% (Pandolfi et al., 2011). The concentrations of $TiO_2$ and $Fe_2O_3$ were indirectly determined on the basis of empirical factors ($TiO_2$ = Ti·0.6 and $Fe_2O_3$ = Fe·0.7)."*

5) P6, L15–19: It is not completely clear what size interval do CoagSDp and NDp represent, and how they were calculated?

**We have made the following changes in section 2.2 for better explanation:**

formed in an NPF event. Thus, the growth rate (GR) is obtained as:

$$"GR_{\Delta D_p} = \frac{dD_p}{dt} = \frac{\Delta D_p}{\Delta t} \qquad\qquad Eq.\ (1)$$

where $D_p$ is the representative diameter of the NPF mode at time t. In this work, the growth rates in the range 4-7 ($GR_{4-7}$) and 7-25 nm ($GR_{7-25}$) were calculated. The uncertainties in the calculated GRs was estimated on 19% and 8% for the 3-7 and 7-20 nm size ranges (Yli-Juuti et al., 2011).

*The formation rate ($J_{D_p}$) is defined as the flux of particles past the lower limit of the size range ($\Delta D_p$), and it is obtained by adding up the observed change in the observed particle number concentration and the losses of particles due to coagulation and growth out of the size range ($\Delta D_p$) and it is calculated following the methodology described by Kulmala et al. (2012):*

$$J_{D_p} = \frac{dN_{\Delta D_p}}{dt} + CoagS_{\Delta D_p} \cdot N_{\Delta D_p} + \frac{GR_{\Delta D_p}}{\Delta D_P} \cdot N_{\Delta D_p} \qquad\qquad Eq.\ (2)$$

where the first term on the right hand side represents the rate of change of particle concentration with time (*where $N_{\Delta D_p}$ is the particle number concentration in the size range $\Delta D_p$*); the second term describes the loss of particles due to coagulation with larger aerosol particles (*where $CoagS_{\Delta D_p}$ is the coagulation sink*); and the third term considers the growth out of the considered size range. *The coagulation sink was calculated from the geometric mean of the considered size range ($\Delta D_p$) to the upper SMPS diameter limit (500 nm), according to Kulmala et al. (2001). In this study, we calculated the formation rates at diameters ($D_P$) 4 nm ($J_4$) and 7 nm ($J_7$) using the diameter range ($\Delta D_p$) of 4-7 and 7-25 nm, respectively."*

6) P8, L18–19: How can you explain the high contribution of the nucleation mode (49%) to the total particle numbers on non-NPF even days?

**This contribution at SNS and UGR sites are for the whole campaign. Despite it is not included in the manuscript, the contribution of nucleation mode aerosol concentrations is lower during non-event days at both measurement sites (~38% at both sites).**

Minor comments:

7) Abstract and P3, L30 and L32: Abbreviation a.s.l. should be resolved.

**Done**

8) P5, L6: Abbreviations RH and PSL should be resolved.

**Done**

9) P8, L32: Rewording is required (in aerosol nucleation mode aerosol concentration).

**Done**

10) P9, L10–11: Repetition; it should be removed.

**We agree and made the following change:**

*"The results of NPF classification (events, non-events or undefined and bad-data days) are summarised in Table 2."*

Another important result is that the difference between GRs at SNS and UGR stations is lower for the small size range, being 53% and 14% for the size range of 7-25 and 4-7 nm, respectively. It is important to note that, although it is not a direct method, the GR at lowest diameter ranges is a good indicator of the quantity of the condensable gases that contribute to the early growth

20  of new particles. Thus, this result suggests that the initial steps of the aerosol formation have almost similar precursors and availability at both sites, resulting in only 14% relative difference of GR at 4-7 nm range. Also, the results suggest higher differences of the available gas precursors for further steps of the aerosol growth, increasing the differences in GRs from 14% in 4-7 nm to 53% in 7-25 nm range. Also, the increase of GR as a function of the particle size suggests that the growth of the particles is not only due to the same vapours that form particles through nucleation.

25  However, due to the difference of altitude, the differences in temperature, RH and UV-B between both stations could contribute, at least partly, to this difference in GRs between SNS and UGR. In this sense, lower temperatures at SNS than at UGR (mean temperatures of 14 and 29 ºC during event days, respectively) can decrease the evaporation rates, enhancing the effective condensation and thus particle growth at lower temperatures. $H_2SO_4$ is produced from the $SO_2 + OH \rightarrow SO_3$ reaction and $OH$ radicals are produced from water vapour UV absorption. Thus, higher RH and UV-B radiation at SNS (mean values

30  of 44% and 2.6 W m$^{-2}$ at SNS and 21% and 2.3 W m$^{-2}$ at UGR) can increase the $H_2SO_4$-water nucleation and $H_2SO_4$ production, respectively, and thus enhance particle growth, especially in the initial NPF steps. Also, as Boulon et al. (2011) and Manninen

et al. (2010) pointed, ion-mediated nucleation could be promoted at higher altitudes compared to low altitudes, and therefore can contribute to the observed differences. However, as Lehtipalo et al. (2016) showed, this mechanism only accelerates the growth of newly formed particles at low concentrations of base compounds. When a strongly basic compound is present, the growth of newly formed particles can be greatly enhanced by acid-base clusters. Thus, the presence of stabilizing vapours could also be responsible of the differences observed.

[revised manuscript text omitted]

30  cm$^{-3}$) was 3 times higher than on 24 June ($1.3 \times 10^7$ molec cm$^{-3}$). As discussed before, the estimated concentrations of sulfuric acid could only explain less than 10% of the observed GRs. These results evidence that sulfuric acid and CS don't play a relevant role in the particle growth during the NPF events observed during these dusty days. However, the significantly high

VOCs concentration observed on 24 June (8.4 µg m$^{-3}$) as compared to 26 June (5.4 µg m$^{-3}$) can explain at least partly the differences observed on GR from 24 to 26 June.

Furthermore, the results suggest that sulfuric acid is not a limiting factor for the occurrence of NPF event on 25 June dusty day, since an increase on sulfuric acid concentrations is observed from 24 to 25 June (Fig. 9a). However, as can be seen in Fig. 9a, VOCs concentration showed a decrease by ~67% from 24 to 25 June. The significant reduction of VOCs concentrations on 25 June could be a possible factor limiting the occurrence of NPF on this dusty day. Thus, the results point to VOCs as one of the main driving factors controlling the occurrence of NPF event and subsequent particle growth at SNS site during Saharan dust events.

*[Figure 9]*

Finally, the chemical composition of desert dust particles (acting as CS) may be another important factor that 
[revised manuscript text omitted]

Leaitch, W. R., Korolev, A., Aliabadi, A. A., Burkart, J., Willis, M. D., Abbatt, J. P. D., Bozem, H., Hoor, P., Köllner, F., Schneider, J., Herber, A., Konrad, C. and Brauner, R.: Effects of 20-100nm particles on liquid clouds in the clean summertime Arctic, Atmos. Chem. Phys., doi:10.5194/acp-16-11107-2016, 2016.

Lehtipalo, K., Rondo, L., Kontkanen, J., Schobesberger, S., Jokinen, T., Sarnela, N., Kürten, A., Ehrhart, S., Franchin, A., Nieminen, T., Riccobono, F., Sipilä, M., Yli-Juuti, T., Duplissy, J., Adamov, A., Ahlm, L., Almeida, J., Amorim, A., Bianchi, F., Breitenlechner, M., Dommen, J., Downard, A. J., Dunne, E. M., Flagan, R. C., Guida, R., Hakala, J., Hansel, A., Jud, W., Kangasluoma, J., Kerminen, V. M., Keskinen, H., Kim, J., Kirkby, J., Kupc, A., Kupiainen-

Määttä, O., Laaksonen, A., Lawler, M. J., Leiminger, M., Mathot, S., Olenius, T., Ortega, I. K., Onnela, A., Petäjä, T., Praplan, A., Rissanen, M. P., Ruuskanen, T., Santos, F. D., Schallhart, S., Schnitzhofer, R., Simon, M., Smith, J. N., Tröstl, J., Tsagkogeorgas, G., Tomé, A., Vaattovaara, P., Vehkamäki, H., Vrtala, A. E., Wagner, P. E., Williamson, C., Wimmer, D., Winkler, P. M., Virtanen, A., Donahue, N. M., Carslaw, K. S., Baltensperger, U., Riipinen, I., Curtius, J., Worsnop, D. R. and Kulmala, M.: The effect of acid-base clustering and ions on the growth of atmospheric nano-particles, Nat. Commun., doi:10.1038/ncomms11594, 2016.

Leng, C., Zhang, Q., Tao, J., Zhang, H., Zhang, D., Xu, C., Li, X., Kong, L., Cheng, T., Zhang, R., Yang, X., Chen, J., Qiao, L., Lou, S., Wang, H. and Chen, C.: Impacts of new particle formation on aerosol Cloud Condensation Nuclei (CCN) activity in Shanghai: Case study, Atmos. Chem. Phys., doi:10.5194/acp-14-11353-2014, 2014.

[revised manuscript text omitted]